# DNA methylation at enhancers identifies distinct breast cancer lineages

Thomas Fleischer[1], Xavier Tekpli[1,2], Anthony Mathelier [1,3], Shixiong Wang[3], Daniel Nebdal[1], Hari P. Dhakal[4], Kristine Kleivi Sahlberg[1,5], Ellen Schlichting[6], Oslo Breast Cancer Research Consortium (OSBREAC), Anne-Lise Børresen-Dale[1], Elin Borgen[4], Bjørn Naume[6], Ragnhild Eskeland[7,8], Arnoldo Frigessi[9], Jörg Tost [10], Antoni Hurtado[1,3] & Vessela N. Kristensen[1,2,11]

Breast cancers exhibit genome-wide aberrant DNA methylation patterns. To investigate how these affect the transcriptome and which changes are linked to transformation or progression, we apply genome-wide expression–methylation quantitative trait loci (emQTL) analysis between DNA methylation and gene expression. On a whole genome scale, in *cis* and in *trans*, DNA methylation and gene expression have remarkably and reproducibly conserved patterns of association in three breast cancer cohorts ($n = 104$, $n = 253$ and $n = 277$). The expression–methylation quantitative trait loci associations form two main clusters; one relates to tumor infiltrating immune cell signatures and the other to estrogen receptor signaling. In the estrogen related cluster, using ChromHMM segmentation and transcription factor chromatin immunoprecipitation sequencing data, we identify transcriptional networks regulated in a cell lineage-specific manner by DNA methylation at enhancers. These networks are strongly dominated by ERα, FOXA1 or GATA3 and their targets were functionally validated using knockdown by small interfering RNA or GRO-seq analysis after transcriptional stimulation with estrogen.

[1] Department of Cancer Genetics, Institute for Cancer Research, Oslo University Hospital, The Norwegian Radium Hospital, Oslo, Norway. [2] Department of Clinical Molecular Biology and Laboratory Science (EpiGen), Akershus University hospital, Division of Medicine, Lørenskog, Norway. [3] Centre for Molecular Medicine Norway (NCMM), University of Oslo, Oslo, Norway. [4] Department of Pathology, Oslo University Hospital, Oslo, Norway. [5] Department of Research, Vestre Viken Hospital Trust, Drammen, Norway. [6] Department of Oncology, Oslo University Hospital, Oslo, Norway. [7] Department of Biosciences, University of Oslo, Oslo, Norway. [8] Department of Immunology, Norwegian Center for Stem Cell Research, Oslo University Hospital, Oslo, Norway. [9] Department of Biostatistics, Oslo Centre for Biostatistics and Epidemiology, University of Oslo and Research Support Services, Oslo University Hospital, Oslo, Norway. [10] Laboratory for Epigenetics and Environment, Centre National de Génotypage, CEA–Institut de Génomique, Evry, France. [11] Institute of Clinical Medicine, Faculty of Medicine, University of Oslo, Oslo, Norway. Thomas Fleischer and Xavier Tekpli contributed equally to this work. Correspondence and requests for materials should be addressed to V.N.K. (email: Vessela.N.Kristensen@rr-research.no)
A full list of consortium members appears at the end of the paper

Alterations in DNA methylation patterns are considered to be an early event in tumor development[1] and have emerged as a hallmark of many cancer types including breast cancer[2]. Aberrant DNA methylation has been frequently associated with clinical and histopathological features of breast cancer patients, such as tumor stage, hormone receptor status, survival time, or somatic mutations, as well as molecular (PAM50) subtypes[3–8]. Noteworthy, estrogen receptor (ER)-positive breast tumors display more pronounced changes in their DNA methylation landscape compared to adjacent normal tissue than ER-negative tumors[3, 6]. However, it is still unclear how such genome-wide DNA methylation alterations explain breast cancer heterogeneity.

Transcription factors (TFs) are key proteins involved in the regulation of gene transcription. They specifically bind to the DNA at cis-regulatory regions local (promoters) or distal (enhancers) to the transcription start sites (TSSs)[9, 10]. Functional genomics experiments mapping TF binding sites confirmed the importance of enhancer activity in regulating transcription[11, 12]. In breast cancer, ERα[13, 14], FOXA1[13, 15], and GATA3[16] are three TFs contributing to the regulation of genes associated with estrogen dependent tumor growth. A recent unsupervised study of DNA methylation in human cells revealed that enhancer regions are differentially methylated in a cell-type-specific manner[17]. However, how DNA methylation at enhancers and transcription factor binding sites affects breast cancer pathogenesis is still poorly understood.

Aberrant CpG methylation at cis-regulatory regions-like promoters is often associated with repression of the associated gene[18]. However, a large portion of aberrantly methylated CpGs in breast cancer are located in intergenic regions[7]. One of our recent studies demonstrated that methylation of CpGs as far as 100 kb away from the TSS of a gene could be associated with its expression[5]. Therefore, CpGs in intergenic or enhancer regions may play an important role in the development of breast cancers through the regulation of the expression of distant genes.

We apply for the first time genome-wide expression–methylation quantitative trait loci (emQTL) analysis between DNA methylation and gene expression, and discover that DNA methylation at enhancers and ERα, FOXA1, and GATA3 binding regions is a breast cancer subtype-specific phenotypic feature. Our results reveal a hitherto unknown connection between the epigenome, TF binding activity, and gene expression in breast cancer. Our analysis also highlights the link between tumor-infiltrating immune cells and the cancer cell epigenome.

## Results

**Identification and validation of 5meCpG–gene pairs by emQTL.** Significant correlations between the level of DNA methylation at a CpG site and gene expression were investigated in a breast cancer discovery cohort with matching genome-wide expression and DNA methylation data (MicMa, $n = 104$). For CpGs with interquartile range >0.1 ($n = 189,026$) and genes ($n = 17,558$), all possible Pearson's correlations between 5meCpG and gene expression were tested for non-zero correlation. We identified 1,115,448 significant CpG–gene pairs (Bonferroni corrected $p$-value < 0.05), of which 739,608 were validated in an independent patient cohort with matching DNA methylation and expression data (TCGA, $n = 253$). Of the non-validated 375,840 emQTL pairs in TCGA, 298,214 could not be tested due to either missing methylation or expression values. Therefore, we found that 90% of the testable MicMa-emQTLs were validated in TCGA, which underlined that the observed correlations between DNA methylation and gene expression were highly conserved across the two cohorts. The validated associations involved the expression of 2664 genes and the methylation at 27,561 CpGs. As

our analysis was not restricted to any distance parameter, CpGs were associated to the expression of genes in cis (same chromosome) or in trans (different chromosome). A significant association between methylation and expression (CpG-gene pair) is hereafter referred to as a expression-methylation Quantitative Trait Loci (emQTL; see flowchart in Supplementary Fig. 1).

To further confirm the observed associations between DNA methylation and gene expression, DNA methylation profiles were generated for a new breast cancer cohort (OSL2, $n = 330$). This data set of DNA methylation is available in GEO with accession number GSE84207. The DNA methylation data was matched to previously published expression data (GEO accession number GSE58215[19]) to obtain 277 samples with matching DNA methylation and gene expression. We performed the emQTL analysis ab initio (i.e., tested all possible CpG–gene pairs) and we rediscovered 95.5% of the total emQTL identified from the MicMa and TCGA cohorts. The observed associations (emQTLs) were conserved between three independent breast cancer cohorts.

**Identification of emQTL clusters.** To elucidate the biological relevance of the emQTL, we performed unsupervised clustering of the Bonferroni corrected $p$-values of each 5meCpG–gene expression pairs. Two very strong and distinct bi-clusters (cliques) of CpGs and genes became apparent: Cluster 1 (3401 CpGs and 160 genes) and Cluster 2 (3601 CpGs and 270 genes) (Fig. 1a).

Gene set enrichment analysis using the Molecular Signatures Database v4.0 (MSigDB[20]) indicated that genes in Cluster 1 were enriched in processes related to the immune system, while genes in Cluster 2 were associated with estrogen response (Fig. 1b–c). The high degree of absolute co-methylation and co-expression of CpGs and genes in Cluster 1 and 2 further highlighted the common regulatory role shared by the CpG-gene pairs in each cluster (Fig. 1d).

**emQTL-CpGs are enriched at enhancers and TF binding regions.** In order to investigate how these clusters of emQTLs, with distinct biological functions, occur we further sought for common transcriptional networks that may explain them.

*Functional genomic location of the emQTL-CpGs.* First, we characterized the functional genomic location of CpGs in emQTL using the ChromHMM segmentation of the human genome in the breast cancer cell line MCF7[21]. Enrichment in a functional region was measured as a ratio between the frequency of emQTL-CpGs found in a specific segment type over the expected frequency of CpGs from the Illumina HumanMethylation450 array (Fig. 2a) or all hg19-CpGs (Supplementary Fig. 2A). We found CpGs in emQTLs significantly enriched in predicted enhancer regions (hypergeometric test $p$-value $< 1 \times 10^{-10}$). We investigated whether emQTL-CpGs were enriched at MCF7 super-enhancers[22, 23] and found emQTL and Cluster 2 CpGs significantly enriched at super-enhancers using hypergeometric test and the Illumina HumanMethylation450 as background ($p$-values = 1.26e-20 and 3.31e-11 respectively). Cluster 1-CpGs were not significantly enriched at super-enhancers. Notably, the super-enhancer containing the highest number of emQTL-CpGs encompassed the GATA3 gene and ERα binding regions, which highlighted the interrelationship between enhancers and TF-binding activity.

Enhancers are known to carry sequences (motifs) recognized by cell-type specific TFs[10]. We therefore sought for motifs enriched in the vicinity (±200 bp) of emQTL-CpGs. We found distinct sets of motifs around the CpGs of Cluster 1 versus Cluster 2 (Supplementary Tables 1 and 2). The most significantly enriched motifs in Cluster 1 were associated with TFs involved

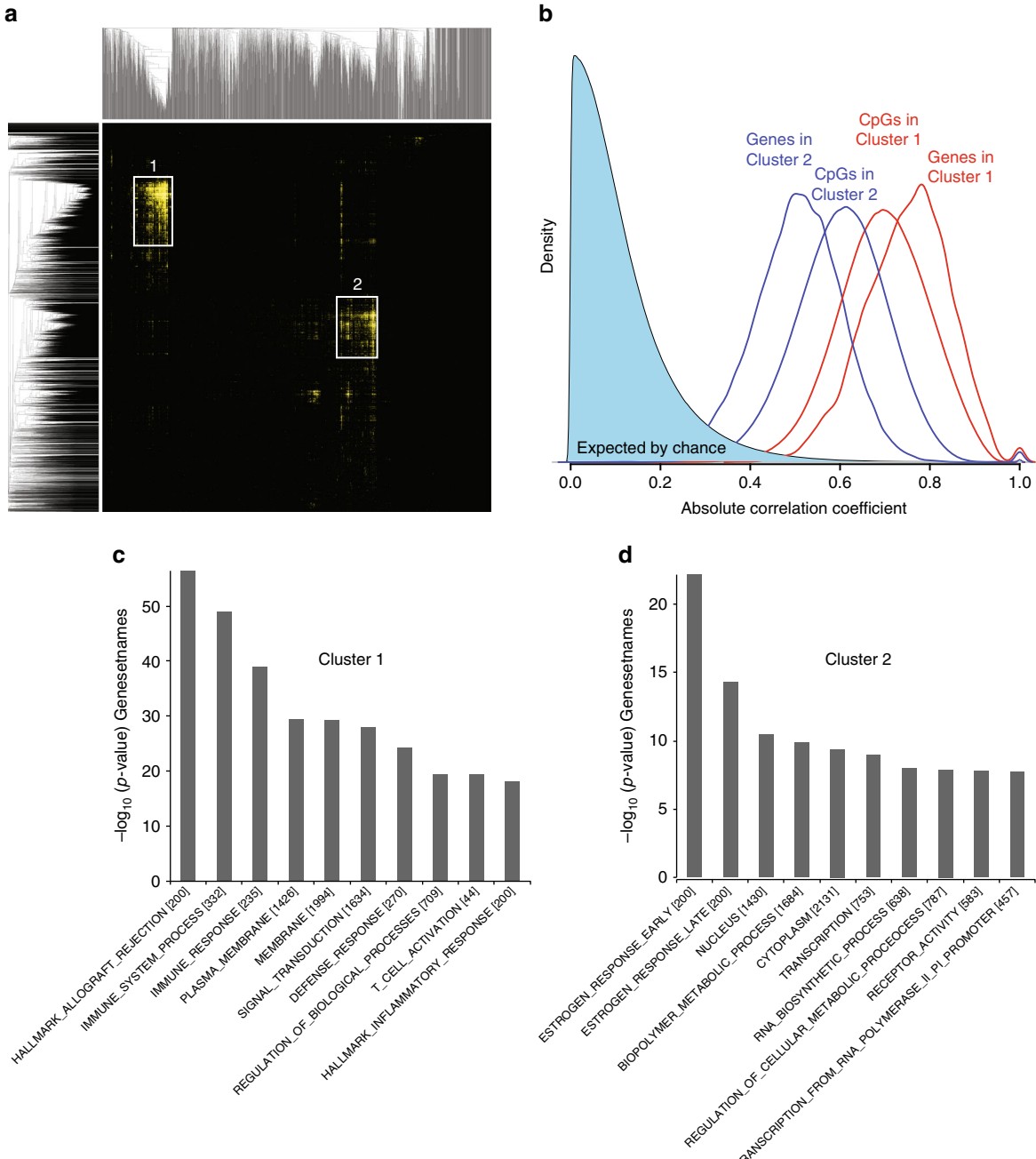

**Fig. 1** Identification of expression-methylation QTL (emQTL) **a** Unsupervised clustering of (−log) p-values of the emQTL by Pearson's correlation and average linkage revealed two main clusters of CpG-gene pairs. Rows represent CpGs and columns represents genes. *Yellow* and *light yellow* spots show highly significant associations between CpG methylation and gene expression. **b** Density plot showing the degree of absolute co-expression of genes and co-methylation of CpGs in Cluster 1 (*red*) and Cluster 2 (*blue*). **c, d** Gene set enrichment analysis in Cluster 1 (**c**, n = 160) and Cluster 2 (**d**, n = 270) using MSigDB (H and C5 databases). The height of the *bars* represents the level of enrichment measured as a ratio between the number of genes overlapping an MSigDB H or C5 gene set over the expected frequency if such overlaps were to occur at random

in immune cell homeostasis such as RUNX1[24], FLI1[25] and ERG[26, 27]. While sequences surrounding CpGs of Cluster 2 carried motifs associated with FOXA1 and GATA3, two TFs playing a key role in breast cancer pathogenesis[15, 16].

*Enrichment of emQTL-CpGs at TF binding regions.* We further screened experimentally defined TF binding regions using 689 uniformly processed human ChIP-seq data sets from ENCODE[28]. We found enrichment of TF-binding regions from blood cells derived ChIP-seq data sets around the CpGs (±200 bp) defining Cluster 1. The TFs were involved in immune processes such as RUNX[24] or PU.1[29]. The CpG regions defining Cluster 2

overlapped significantly with FOXA1 and GATA3 binding regions (ChIP-seq peaks) from breast cancer cell line experiments (Supplementary Tables 3 and 4). These analyses of CpG regions recapitulated the gene set enrichment analysis and emphasized the distinct biological functions of Clusters 1 and 2.

Next, we focused on the TF binding regions obtained from the MCF7 cell line and mined 71 ChIP-seq experiments from ENCODE[28] and 40 available at GEO retrieved from ReMap[30]; we performed enrichment analyses using hypergeometric tests. We observed that emQTL-CpGs were most strongly enriched at ERα, FOXA1, and GATA3 binding sites and to a lesser extend at

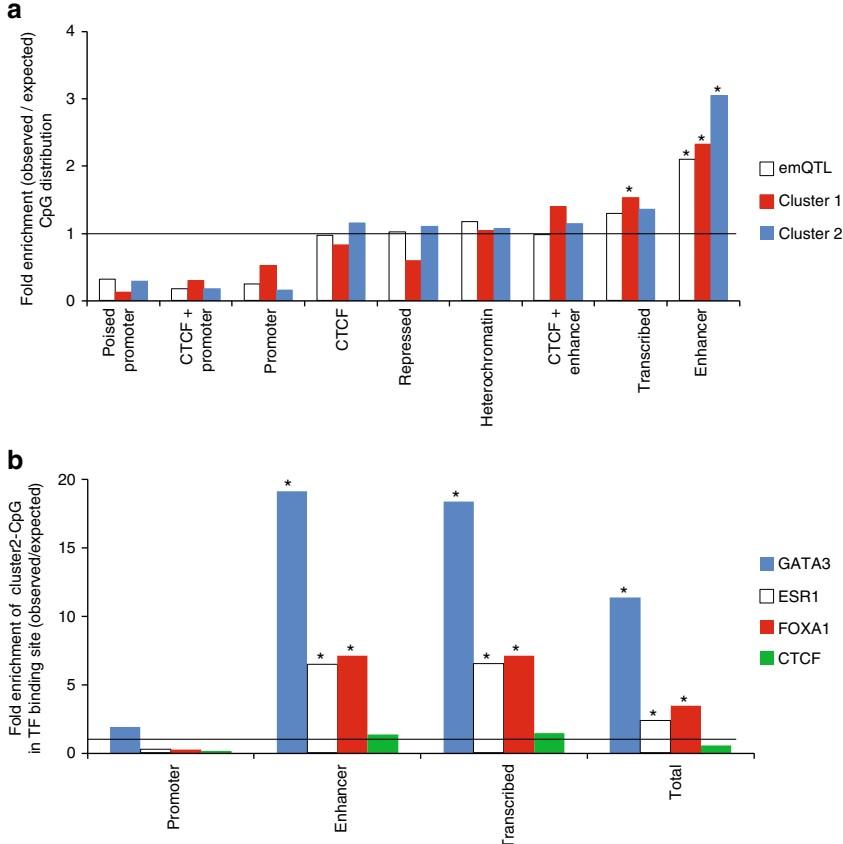

**Fig. 2** Genomic location of emQTL-CpGs according to ChromHMM and TF binding regions **a** Bar plot showing the enrichment of emQTL, Cluster 1 and Cluster 2 CpGs over the expected frequency of CpGs from the Illumina HumanMethylation450 across functional/regulatory regions of the genome as determined by MCF7 ChromHMM annotation[62]. The height of the *bars* represents the level of enrichment measured as a ratio between the frequency of all emQTL-CpGs (*white*), CpGs in Cluster 1 (*red*) and CpGs in Cluster 2 (*blue*) overlapping a functional element over the expected frequency if such overlaps were to occur at random. Statistically significant enrichments ($p < 0.05$; hypergeometric test) are marked with an asterisk. **b** Enrichment analysis of CpGs in Cluster 2 across ERα (*white*), GATA3 (*blue*), FOXA1 (*red*) and CTCF (*green*) binding regions as determined by ChIP-seq analysis. Enrichment is calculated for different genomic regions determined by MCF7 ChromHMM annotation. For this analysis some ChromHMM annotations were collapsed into one as follow: Enhancer = 'Enhancer' and 'Enhancer + CTCF' and Promoter = 'Promoter', 'Promoter + CTCF' and 'Poised Promoter'. The height of the bars represents the level of enrichment measured as a ratio between the frequencies of CpGs in Cluster 2 in each ChIP-seq peak at specific regulatory region over the expected frequency if such overlaps were to occur at random. Statistically significant ($p < 0.05$; hypergeometric test) enrichments are marked with an asterisk

GREB1, FOS, DPF2, AHR, and ZNF217-binding regions (Supplementary Fig. 2B and Supplementary Data 1). To further assess the enrichment of Cluster 2 CpGs at TF binding regions, we performed a permutation test (1000 permutations), in which we randomly selected 3601 CpGs from the 450k array (number of CpGs in Cluster 2) and calculated how many were in either ERα[31], GATA3[16], or FOXA1[15, 16] binding regions. None of the randomly selected combinations of CpGs showed a similar enrichment as the CpGs in Cluster 2 (*p*-value < 0.001).

We further investigated in which ChromHMM genomic annotations Cluster 2 CpGs at FOXA1[15, 16], ERα[31], GATA3[16], and CTCF (as control) binding regions were located. The most pronounced enrichment in TF binding regions was for CpGs in Cluster 2 at ChromHMM predicted enhancers when compared to the distribution of CpGs on the HumanMethylation450 array (Fig. 2b) or all CpGs (hg19; Supplementary Fig. 2C). The hg19 locations of the Cluster 2 CpGs, their MCF7 ChromHMM segmentation annotations, and TF binding regions are provided in Supplementary Data 2.

All together, these results clearly show that CpGs in Cluster 2 are mainly found at enhancers containing FOXA1, GATA3, and ERα-binding regions.

**DNA methylation of CpGs in the estrogen-signaling emQTL**. Having found that genes in Cluster 2 were associated with estrogen signaling, and that CpGs in Cluster 2 were located in binding regions of ERα, FOXA1, and GATA3, we further investigated the level of DNA methylation of these CpGs in regard to histopathological features and molecular classification of breast cancer patients. We performed unsupervised clustering (Fig. 3a) based on the level of DNA methylation of CpGs in Cluster 2 (n = 3601) of breast tumor samples from TCGA (n = 609). The level of CpG methylation in Cluster 2 clearly distinguished between ER positive and negative breast cancers (Fig. 3a). In addition, CpGs were clearly separated in two sub-clusters: CpG-Cluster 2A and CpG-Cluster 2B. CpGs in Cluster 2A were mainly found in binding regions of ERα, FOXA1, and GATA3, and showed lower methylation in ER positive compared to ER-negative breast tumors. CpGs in Cluster 2B showed an inverse pattern of DNA methylation, i.e., higher methylation in ER positive compared to ER-negative tumors (Fig. 3a).

To further validate whether the specific DNA methylation patterns in Cluster 2 were distinguishing patients according to ER status and CpG locations according to TF binding regions, we performed unsupervised clustering of the CpGs in Cluster 2 of

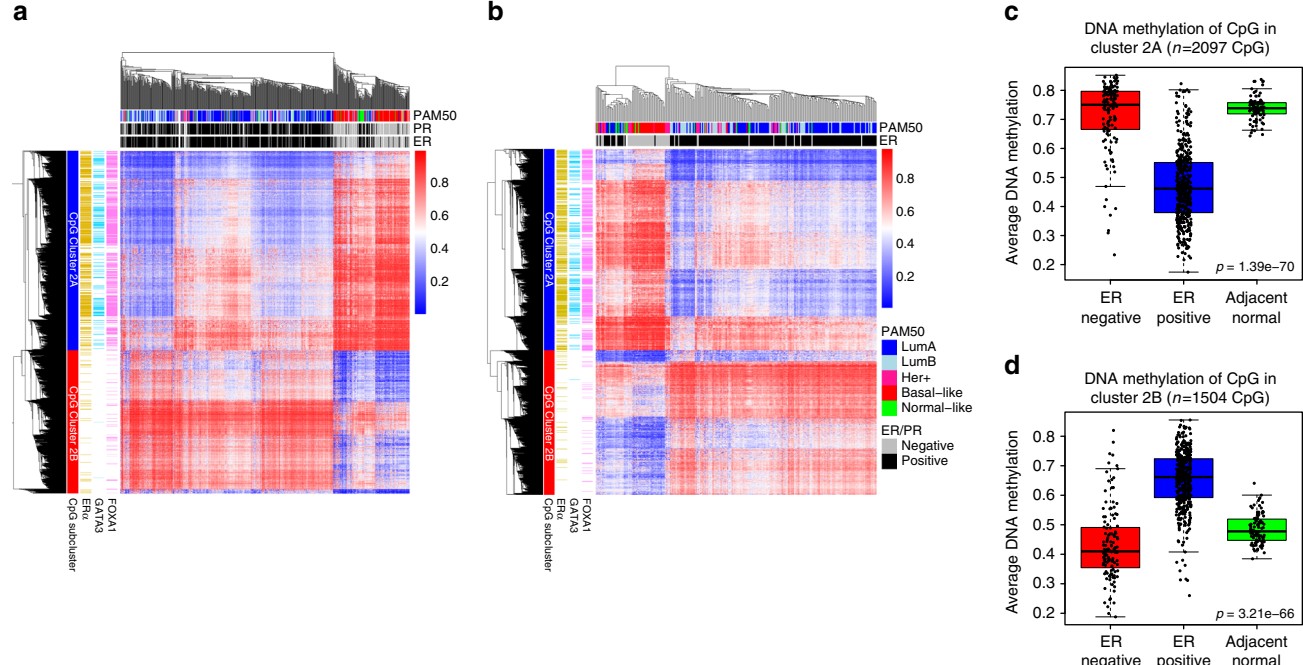

**Fig. 3** Differential DNA methylation of Cluster 2 CpGs in TF binding regions. Unsupervised clustering of DNA methylation levels of the CpGs in Cluster 2 for (**a**) the TCGA cohort (*N* = 609) and (**b**) the OSL2 cohort (*N* = 272). Annotation of the rows of the heatmap shows whether a CpG is located in ERα (*yellow*), FOXA1 (*pink*) or GATA3 (*light blue*) binding regions according to ChIP-seq experiments in MCF7 cells. Annotations of the column of the heatmap indicate histopathological features of the patients: PAM50 subtype, ER and PR status. **c–d** Average DNA methylation of **c** the 2097 CpGs in Cluster 2A and D) 1504 CpGs in Cluster 2B for samples in the TCGA cohort. Boxplots represent the average DNA methylation of these CpGs in ER positive (*blue*; *n* = 418), ER-negative tumors (*red*; *n* = 124) and adjacent normal tissue (*green*; *n* = 97). Kruskal-Wallis test *p*-values are denoted

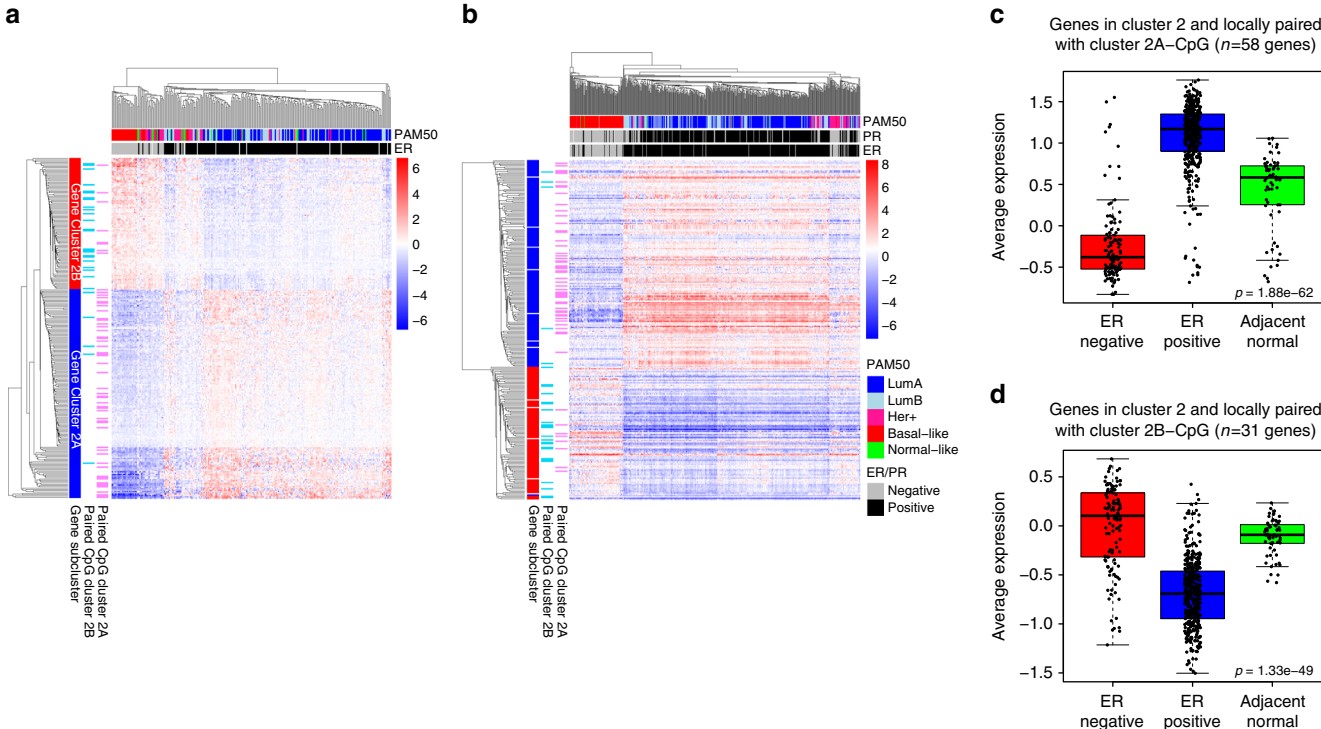

**Fig. 4** Differential expression of genes in Cluster 2. Unsupervised clustering of expression of the genes in Cluster 2 for (**a**) the OSL2 cohort (*N* = 272) and (**b**) the TCGA cohort (*N* = 528). Genes in *rows* are annotated if they are locally paired with CpG in Cluster 2 A (*pink*) or paired with CpG of Cluster 2B (*light blue*). Two main sub-clusters of genes are identified in the heatmap in **a**, and the genes in the heatmap in **b** are annotated correspondingly. Annotations of the column of the heatmap indicate histopathological features of the patients: PAM50 subtype, ER and PR status. **c–d** Average expression of genes in Cluster 2 locally paired with CpGs in Cluster 2 A (*n* = 58) and Cluster 2B (*n* = 31). A gene was considered to be locally paired with a CpG if it was situated not more than 10 kb from this CpG. Average expression in ER positive tumors (*blue*; *n* = 406), ER-negative tumors (*red*; *n* = 117) and adjacent normal tissue (*green*; *n* = 61) from the TCGA cohort. Kruskal–Wallis test *p*-values are denoted

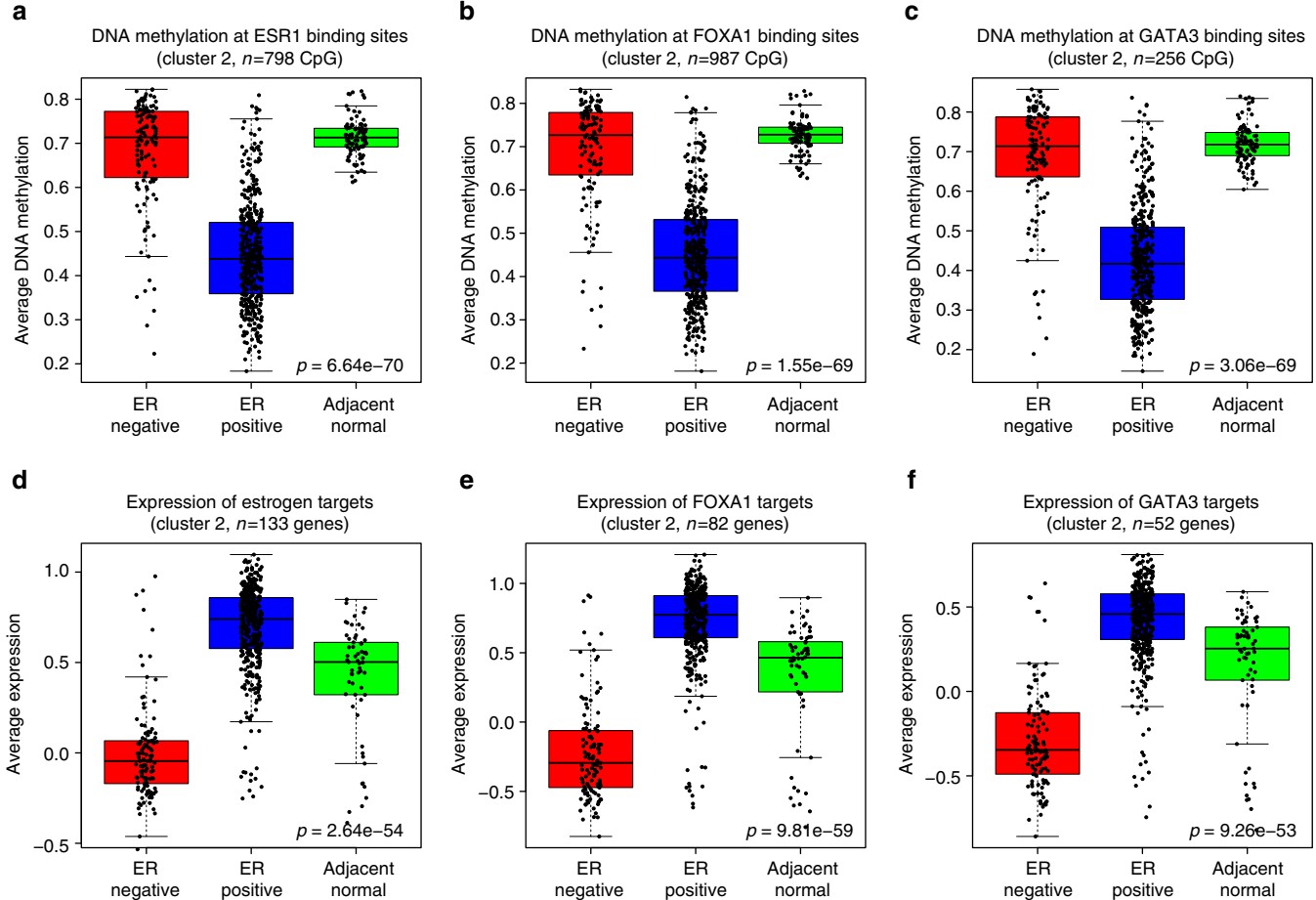

**Fig. 5** DNA methylation of ERα, FOXA1 and GATA3 binding regions and expression of their target genes in Cluster 2. **a–c** Average DNA methylation of CpGs in Cluster 2 and ERα **a**, FOXA1 **b** and GATA3 **c** binding regions defined by ChiP-seq peaks. Boxplots represent the average DNA methylation of these CpGs in ER positive (*blue*, n = 418), ER-negative tumors (*red*, n = 124) and adjacent normal tissue (*green*, n = 97). The average methylation of the Cluster2-CpGs in a TF binding site was significantly lower in ER positive patients compared to ER-negative and adjacent normal tissue. **d–f** Average gene expression of TF target genes in Cluster 2. **d** Estrogen (GRO-seq), **e** FOXA1 (siRNA) and **f** GATA3 (siRNA). Boxplots represent the average expression of the TF target genes in ER positive tumors (*blue*, n = 406), ER-negative tumors (*red*, n = 117) and adjacent normal tissue (*green*, n = 61). The average expression was significantly higher in ER positive tumors compared to ER-negative and adjacent normal tissue. Kruskal–Wallis test *p*-values are denoted

breast tumor samples from OSL2 (272 samples; 3527 CpGs). The results from the TCGA were recapitulated; the subset of CpGs in Cluster 2 and in binding regions of ERα, FOXA1, and GATA3 had clear differences in methylation according to ER status, and allowed separation of patients with ER positive versus ER-negative disease (Fig. 3b).

We further compared the level of DNA methylation in tumor tissues to adjacent normal tissue for CpGs in both Cluster 2A and Cluster 2B (Figs. 3c-d). We found that compared to normal, CpGs in Cluster 2A were specifically hypomethylated in ER positive tumors (Fig. 3c, Kruskal–Wallis test, *p*-value = $1.39 \times 10^{-70}$), while CpGs in Cluster 2B were hypermethylated in ER positive tumors (Fig. 3d, Kruskal–Wallis test, *p*-value = $3.21 \times 10^{-66}$). These results suggest that the specific methylation patterns of CpGs in Cluster 2 are features acquired during carcinogenesis.

**Expression of genes in the estrogen-signaling emQTL cluster.**
The unsupervised clustering of expression of genes in Cluster 2 allowed almost perfect separation of ER positive samples and ER-negative samples in two patient cohorts (OSL2 and TCGA; Fig. 4a, b, respectively). We identified two gene sub-clusters with differential expression according to ER status (Gene-Cluster 2A and Gene-Cluster 2B).

In order to identify the epigenetic regulation of expression strictly in *cis*, we investigated to which extent the genes in Cluster 2 were located nearby (±10 kb window) with CpGs of the same Cluster. We found that 32% of the genes in Cluster 2 were paired locally (i.e. were within 10 kb window) with at least one CpG of the Cluster, suggesting a local regulation of Cluster 2 genes through DNA methylation of enhancers carrying TF binding regions for FOXA1, GATA3 and ERα. The CpGs with low methylation in ER positive disease (Cluster 2A-CpGs) were mainly paired locally with genes with high expression in ER positive patients (Gene-Cluster 2A; Fig. 4c, Kruskal-Wallis test, *p*-value = $1.88 \times 10^{-62}$). *Vice versa*, the CpGs with low methylation in ER-negative disease (Cluster 2B-CpGs) were mainly paired locally with genes with high expression in ER-negative patients (Gene-Cluster 2B; Fig. 4d, Kruskal-Wallis test, *p*-value = $1.33 \times 10^{-49}$).

**DNA methylation at ERα, FOXA1 and GATA3 binding regions.** We found CpGs in Cluster 2 significantly enriched at binding regions for ERα, FOXA1 or GATA3, we investigated the interplay between TF binding, DNA methylation, and expression of the target genes of these TFs. The DNA methylation of ERα, FOXA1, and GATA3 binding regions was lower in ER positive tumors (Fig. 5a-c, Kruskal–Wallis test, *p*-value equal to $6.64 \times 10^{-70}$, $1.55 \times 10^{-69}$ and $3.06 \times 10^{-69}$, respectively). As

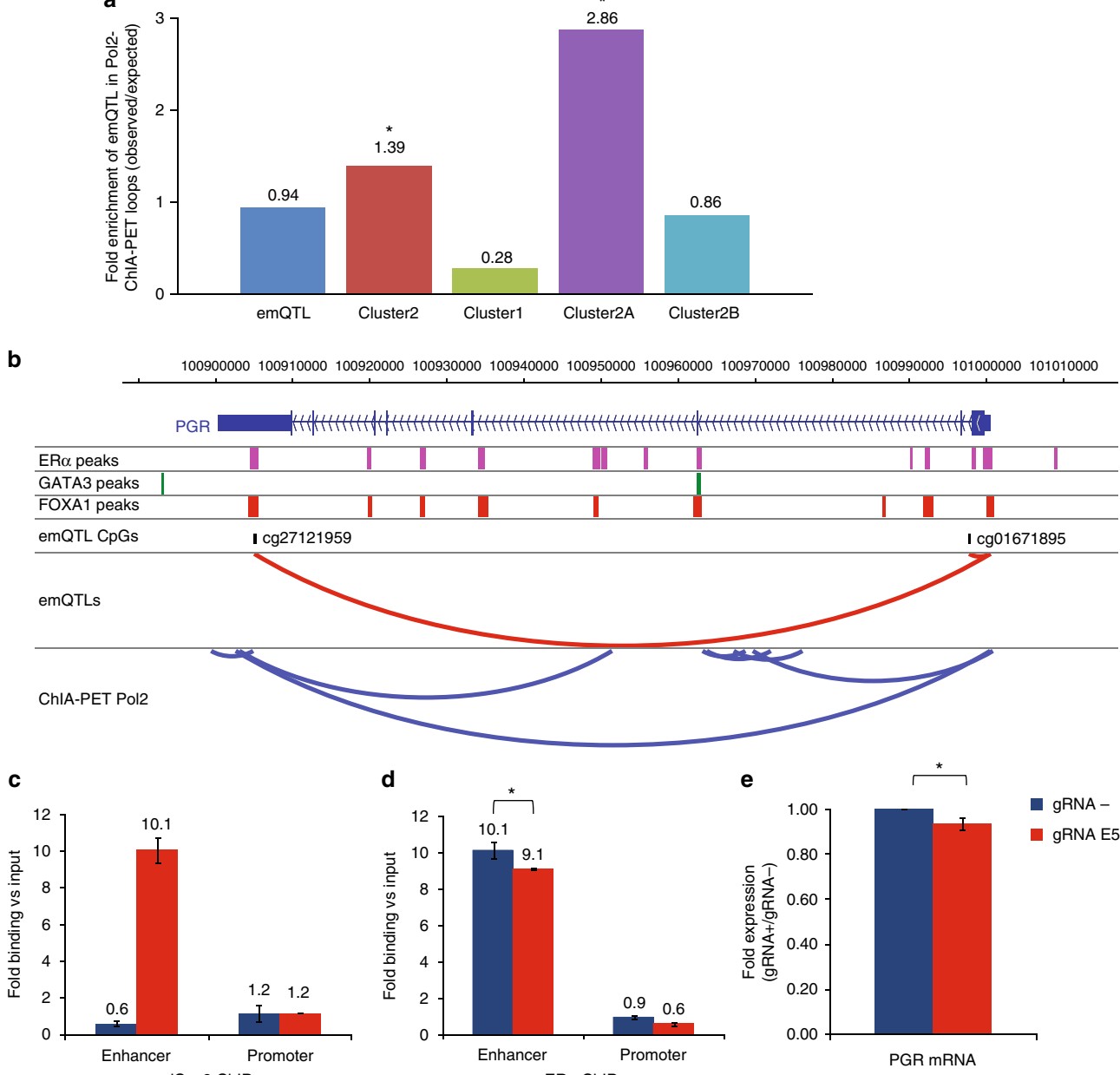

**Fig. 6** Enhancer-promoter interaction and impact of TF binding on target gene expression. **a** Bar plot showing the enrichment of emQTL in ChIA-PET Pol2 loops for Cluster 1, Cluster 2, Cluster 2A and Cluster 2B. The height of the *bars* represents the level of enrichment measured as a ratio between the frequencies of emQTL (CpG–Gene pairs) found in the head and tail of Pol2 loops, over the expected frequency if such overlaps were to occur at random. Statistically significant enrichments (hypergeometric test, *p*-value < 0.05) are marked with an asterisk. **b** Example of overlap of emQTL (*red arcs*) and ChIA-PET Pol2 loops (*blue arcs*). Also shown are the location of ERα, FOXA1 and GATA3 binding regions. **c, d** dCas9 and ERα ChIP were performed in control (−gRNA and dCas9) or transfected MCF7 cells (gRNA E5 and dCas9), to assess the binding of each protein at enhancer or promoter. Statistically significant differences (*t*-test; two tails, *p*-value < 0.05) are marked with an asterisk. The data are presented as mean of three of independent replicates ± s.d. **e** mRNA levels of PGR were measured in control (-gRNA and dCas9) or transfected MCF7 cells (gRNA E5 and dCas9) by real-time PCR. Statistically significant differences (*t*-test; two tails, *p*-value < 0.05) are marked with an asterisk. The data are the mean of three of independent replicates ± s.d

expected, the level of methylation of these CpGs was also lower in Luminal A and Luminal B versus Normal-like and Basal-like breast cancer subtypes (Supplementary Fig. 3A–C). To investigate whether low methylation of CpGs in TF binding regions was a cancer specific feature of ER positive breast cancer, we compared the tumor methylation levels to those of non-cancerous tissues. We found the level of DNA methylation at TF binding regions systematically lower in ER positive tumors when compared to adjacent normal tissue (Fig. 5a–c). As adjacent normal tissue to

tumors does not always reflect the physiologically normal breast[32] we compared the methylation levels between tumors and healthy breast tissue from reduction mammoplasty (GSE60185[5]). We found that 82.5, 84.7, and 87.9% of the CpGs in Cluster 2 and in binding regions of ERα, FOXA1, and GATA3, respectively, were significantly demethylated in ER positive disease when compared to normal tissue (difference in median methylation at least 0.1 (10%) and nominal *p*-value < 0.05; examples in Supplementary Fig. 4A–C).

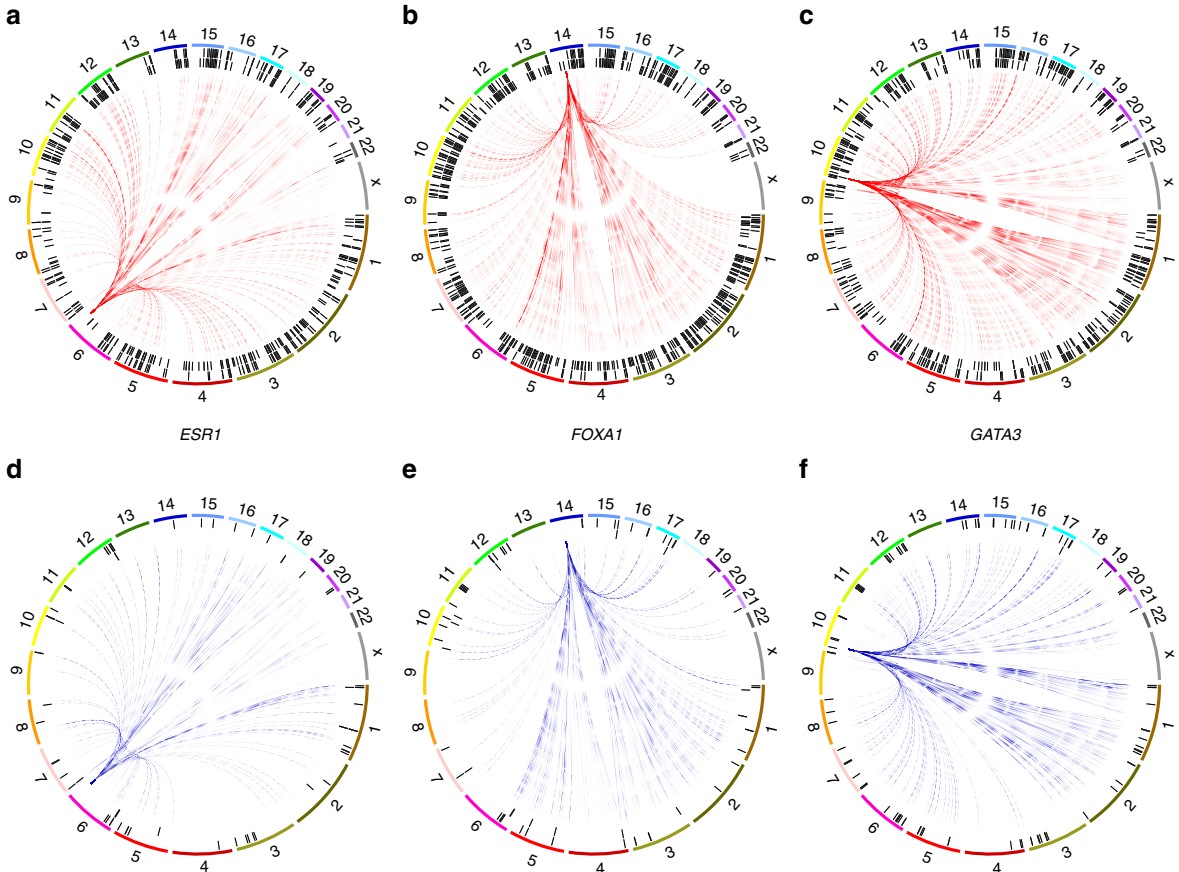

**Fig. 7** Circos plots showing the genomic location of all CpGs in emQTL with *ESR1*, *FOXA1* and *GATA3*. Circos plot representing all associations between CpGs and *ESR1* (**a**, **d**), *FOXA1* (**b**, **e**), and *GATA3* (**c**, **f**). *Red lines* represent negative associations **a–c** and *blue lines* represent positive correlations **d–f**. The outer ring indicates whether a CpGs is located in an enhancer determined by MCF7 ChromHMM annotation and the inner ring whether it is located in a binding region of the respective TF determined by ChIP-seq peaks

**Functional identification of TF target genes**. Targets of FOXA1 and GATA3 were experimentally identified following knockdown of FOXA1 or GATA3 by siRNA in MCF7 cells. Following RNA sequencing of control and knockdown cells, differentially expressed genes were considered either FOXA1 or GATA3 targets. ERα target genes were obtained by analysis of Global Run-On sequencing (GRO-seq[33]), which is a method to identify genes that are being transcribed in MCF7 cells exposed or not to estrogen. Differentially expressed nascent transcripts were assigned target of estrogen and ERα. This set of experiments showed that 67% of the genes in Cluster 2 were targets of ERα, FOXA1, or GATA3. These target genes showed significantly higher expression in ER positive tumors (Fig. 5d–f, Kruskal–Wallis test, *p*-value equal to $2.64 \times 10^{-54}$, $9.81 \times 10^{-59}$ and $9.26 \times 10^{-53}$, respectively). Expression of these target genes was also higher in Luminal A and Luminal B versus Normal-like and Basal-like breast cancer subtypes (Supplementary Fig. 5A–C). We experimentally demonstrated that a large proportion of the genes in Cluster 2 were targets of ERα, FOXA1, or GATA3 confirming the strong link between Cluster 2 and estrogen signaling.

**Functional validation of *cis* emQTL in Pol2 loop**. To further assess the link between DNA methylation at enhancers and the expression of target genes, we used ChIA-PET Pol2 data sets, which enable us to identify experimentally defined long-range chromatin interactions genome-wide through Pol2 binding. We observed that Cluster 2A-CpGs (CpG in ERα, FOXA1 or GATA3 binding regions) in emQTL with Cluster 2A-genes (TF target genes) were significantly enriched at Pol2-ChIA-PET loops (Hypergeometric test, *p*-value < 0.05; Fig. 6a). This provided further evidence for a functional regulation of the expression of the target genes through DNA mehylation of enhancers containing TF binding regions. Overlap between a ChiA-PET Pol2 loop and an emQTL involving a CpG at enhancer and *PGR* expression is shown in Fig. 6b.

We further experimentally assessed whether ERα binding at the enhancer identified in the 3′UTR region of *PGR* was functionally involved in the regulation of *PGR* expression. In MCF7 breast cancer cell line, we inhibited the binding of ERα through the simultaneous expression of dCas9 and a gRNA specifically recognizing *PGR* enhancer. First, we validated the specificity of the gRNA by analyzing the binding of dCas9 by ChIP-PCR at the enhancer (Fig. 6c). The results showed that dCas9 specifically bind at the enhancer and not at the *PGR* promoter. The binding of dCas9 at the targeted enhancer was associated with a significant reduction of ERα binding (10% versus control, *t*-test, *p*-value < 0.05; Fig. 6d) and also with a significant reduction of PGR mRNA expression (7% versus control, *t*-test, *p*-value < 0.05; Fig. 6e).

We identified an overlap between Pol2 loops and in *cis* Cluster 2A-emQTL, and validated the impact on gene expression of TF binding at distal enhancer through an experimental approach using dCas9.

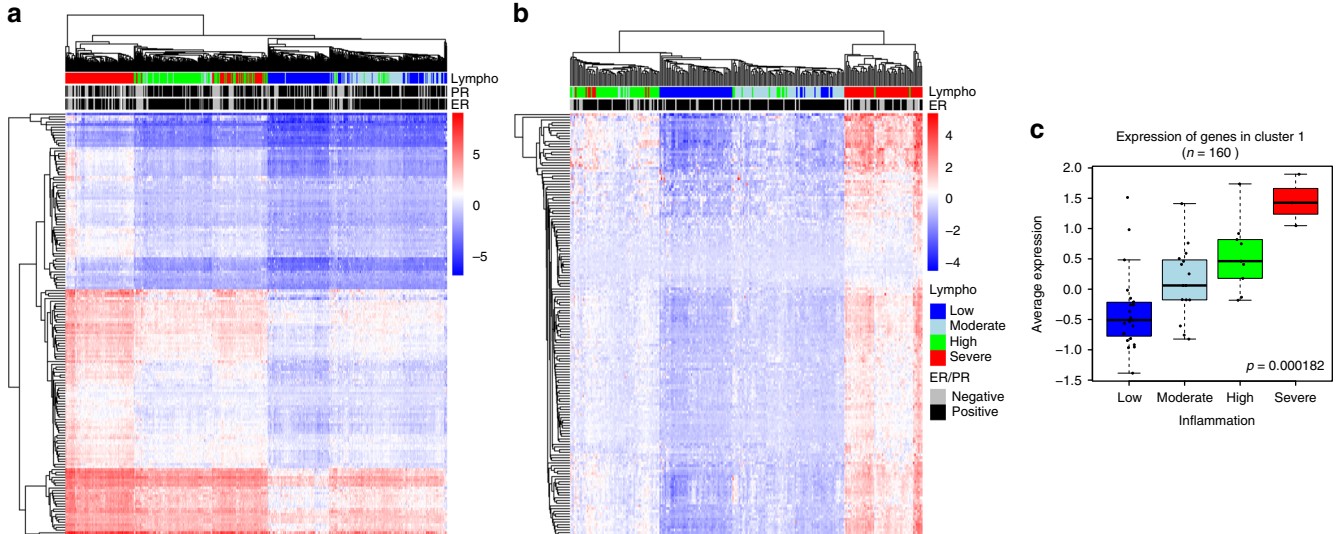

**Fig. 8** Cluster 1 highlights a link between DNA methylation and lymphocyte infiltration. Unsupervised clustering of expression levels of the 160 genes in Cluster 1 from **a** the TCGA cohort ($N = 528$) and **b** the OSL2 cohort ($N = 272$). Annotations of the column indicate the level of lymphocytes infiltration, ER and PR status. Levels of lymphocyte infiltration were calculated from a set of genes expressed by lymphocyte characterized by the algorithm Nanodissect[34]. **c** 68 tumor tissue samples were scored as low ($n = 30$), moderate ($n = 22$), high ($n = 13$) or severe ($n = 3$) inflammation by a pathologist based on the level of immune cell infiltration found in each tumor. Boxplot representing average expression of genes in Cluster 1 ($n = 160$) according to inflammation score. Kruskal–Wallis test $p$-value is denoted

**cis and trans emQTL of ERα, FOXA1 and GATA3 expression**. As stated above, one third of the genes in Cluster 2 were locally paired with the DNA methylation of a CpG *in cis*. Indeed, emQTL analysis identified CpGs in promoters (as predicted by ChromHMM) whose DNA methylation may regulate proximal genes. Prime examples are *ESR1*, *FOXA1*, and *GATA3* themselves, whose expressions were significantly associated (Pearson correlation) with the DNA methylation of CpGs in their promoters (Supplementary Fig. 6; *ESR1*: cg00601836, $r = -0.638$; *FOXA1*: cg27143688, $r = -0.635$; *GATA3*: cg04213746, $r = -0.638$). The DNA methylation of these CpGs showed stronger inverse correlation with the respective gene expression than conventional near-to-TSS regions ($\pm 3$ kb) (*ESR1* TSS, $r = -0.517$; *FOXA1* TSS, $r = -0.300$; *GATA3* TSS, $r = -0.129$). This suggests that emQTL could successfully identify the most prominent *in cis*-regulatory regions whose DNA methylation impact on gene expression.

However, the expression of these three TFs was also in emQTL with CpGs in *trans*. In order to investigate the biological relevance of in *trans* emQTL we further investigated the genomic location of the CpGs in emQTL with these TFs. We found that the CpGs in inverse correlation (negative emQTL) with the expression of *ESR1, FOXA1, or GATA3* were enriched at MCF7 enhancers and at the respective TF binding region (Fig. 7a–c; red circos plots). Conversely, CpG in positive emQTL with *ESR1, FOXA1, or GATA3* were mainly found in heterochromatic regions (Fig. 7d–f; blue circos plots). We have thus revealed a pathway specific inverse relationship between the degree of expression of a TF in breast cancer and the level of methylation of its binding regions genome-wide as an important phenotypic feature distinguishing different breast cancer lineages.

**Cluster 1 reflects tumor infiltration by lymphocytes**. Unlike genes in Cluster 2, genes in Cluster 1 did not segregate breast cancer patients according to PAM50 subtype or ER status. The gene set enrichment analysis (Fig. 1b), the motif (Supplementary Table 1) and the ENCODE ChIP-seq (Supplementary Table 3) enrichment analyses of Cluster 1 clearly indicated that genes and

CpGs in Cluster 1 were associated with immune processes. We further investigated the possible role of immune cell infiltration in the formation of Cluster 1. We used the algorithm Nanodissect[34] to quantify the level of lymphocyte infiltration in TCGA and OSL2 breast cancer samples based on gene expression data. Unsupervised clustering based on the expression of genes in Cluster 1 segregated the patients according to the level of lymphocytes infiltration in the tumor (Fig. 8a: TCGA; 8B: OSL2). To validate this in silico observation, we inspected paraffin embedded tumor tissue of our discovery cohort (MicMa) and scored 68 samples for tumor inflammation based on the quantity of infiltrating immune cells. We found that the expression of the genes in Cluster 1 was highly correlated with the tumor inflammation score (Kruskal-Wallis test, $p$-value = 0.000182; Fig. 8c).

We further characterized the CpGs in Cluster 1, and found either positive (CpG-Cluster 1B) or negative (CpG-Custer 1A) correlations to the expression of genes in Cluster 1 (Supplementary Fig. 7). Cluster 1B-CpGs were often found at ERα, FOXA1, and GATA3 binding regions, which associated with an overall enrichment of CpGs in Cluster 1 at these three TFs binding regions. This enrichment was lower than the one observed in Cluster 2 (Supplementary Fig. 8 and Supplementary Fig. 2B). Average DNA methylation in Cluster 1A and Cluster 1B was highly dependent of lymphocyte infiltration. DNA methylation in Cluster 1A decreased with increasing lymphocyte infiltration, while in Cluster 1B DNA methylation increased with the level of lymphocyte infiltration (Supplementary Fig. 9A, B). We further investigated whether the differential levels of methylation in Cluster 1A and Cluster 1B were related to intratumor heterogeneity and gradual mixture of various cell types in the biopsies. Tumor purity was estimated using InfiniumPurify[35] and 450k methylation data. We found that tumor samples with higher levels of lymphocyte infiltration (Nanodissect) also showed significantly lower tumor purity (Supplementary Fig. 9C). This confirmed that samples with high lymphocyte infiltration, also showed lower percentage of cancer cells. Therefore, DNA methylation signals measured in tumors with high infiltration (low purity) is more susceptible to be influenced by a mixture of

cancer cells, adjacent normal tissue, stroma, and infiltrating immune cells.

Cluster1-CpG DNA methylation of breast cancer cell lines (GSE94943) resembled the methylation of tumors with low infiltration. Tumors with higher lymphocyte infiltration showed intermediate methylation levels between breast cancer cell lines and immune cells. This holds true for both Cluster 1A and Cluster 1B and supports that the observed methylation level in tumors with higher infiltration may be attributed to higher numbers of immune cells. In fact, T cells (GSE79144[36]), B cells (GSE68456[37]), leukocytes (GSE69270[38]), and monocytes (GSE68456[37]) all show low methylation of CpGs in Cluster 1A and high DNA methylation of CpGs in Cluster 1B, oppositely of the methylation level of breast cancer cell lines (Supplementary Fig. 9A, B).

**Subtype-specific emQTL analysis.** From the data presented above it is evident that the emQTLs in Cluster 2 (the estrogen signaling cluster) are at large a result of the reciprocal methylation/expression pattern in ER positive (Luminal) compared to ER-negative (Basal) tumors. To confirm this we performed the emQTL analysis for each PAM50 subtype separately. To be able to retrieve sufficient samples with matching DNA methylation and expression data for each subtypes, we used RNA-seq data from the TCGA cohort (Luminal A: $n = 236$; Luminal B: $n = 137$; Her2 enriched: $n = 49$; Basal-like: $n = 92$; Normal-like: $n = 44$). We identified more associations for Luminal A (4,689,557), Luminal B (1,647,259), and Basal-like (2,036,164) subtypes, than for Her2 enriched (21,161) and Normal-like (178) subtypes. Performing the analysis within each PAM50 subtype, we were able to identify approximately 50% of the Cluster 2-emQTL genes in Luminal A and Luminal B and less than 6% in the other subtypes. Cluster 1 was on the other hand rediscovered in all subtypes (except Normal-like), identifying more than 95% of the Cluster 1-genes in Luminal A, Luminal B, and Basal-like; and 60% in Her2 enriched. These data are shown in Supplementary Table 5.

## Discussion
Through identification of all possible Pearson's correlations between DNA methylation and expression in breast cancer, without restriction on distance or chromosome location, we identified two very distinct key gene regulatory networks known to be involved in breast cancer pathogenesis. The first gene regulatory network was wired by ERα, FOXA1, and GATA3 through an intricate relationship between DNA methylation of their binding regions at enhancers and the expression of their target genes. The second gene regulatory network was related to immune infiltration. When performing emQTL analysis in a subtype-specific manner we identified Cluster 1 within each subtype, however Cluster 2 was only fully discovered when all subtypes were included. These observations are in line with the fact that Cluster 1 did not recapitulate the PAM50 classification and is related to immune infiltration and intratumoral heterogeneity. Conversely, in Cluster 2, the intertumoral heterogeneity between ER positive and ER-negative tumors is the cause for the observed associations between DNA methylation and gene expression. Indeed, Cluster 2 was more strongly associated with PAM50 classification and pinpoints enhancers differentially methylated between ER-positive and ER-negative tumors, containing ERα, GATA3, and FOXA1 binding regions.

A recent study showed that DNA methylation at enhancers and ERα binding sites may participate in ER positive breast cancer resistance to anti-estrogen treatment[39], and in addition, ERα binding regions have been suggested to be altered by DNA

methylation with effects on gene expression[40]. These studies underline the importance of DNA methylation at enhancers and TF binding regions in breast cancer pathogenesis. Our study clearly identified regulatory regions whose methylation status associate with different breast cancer lineages. Cluster 2A contained CpGs in binding regions for ERα, FOXA1, and GATA3 with significantly lower DNA methylation in ER positive compared to ER-negative breast cancer. This may associate with the activity of these specific TFs and estrogen dependent tumor growth. We also demonstrated that enhancers bearing binding regions of ERα, FOXA1, and GATA3 were demethylated in ER positive tumors when compared to adjacent normal tissue and healthy breast tissue (reduction mammoplasty). The epigenetic regulation of these regions may be an early event during normal breast cell transformation into estrogen dependent tumor cells or reflect the DNA methylation of the cell from which the tumor originate[41].

emQTL-CpGs were significantly enriched in distal regulatory regions often overlapping with ChromHMM-MCF7 enhancers and super-enhancers[22, 23]. These enhancers may be representative of ER positive breast cancer enhancers; however tumors may harbor very different biology. As CpGs in Cluster 2A are enriched in MCF7 enhancers, super-enhancers and ERα, FOXA1, and GATA3 binding regions, they might represent conserved distal regulatory regions across ER positive disease. In addition to being putative fundamental drivers of ER positive carcinogenesis, DNA methylation levels at key TF binding sites or enhancers constitute interesting regions for further investigation as predictive and prognostic biomarkers in breast cancer. The role of genetic polymorphisms (SNPs) in further modulating emQTLs and thus contributing to susceptibility and cancer progression is of utmost interest for further studies.

The CpGs in binding regions of ERα, FOXA1 and GATA3 were grouped in CpG-Cluster 2A, while the CpGs in Cluster 2B showed an inverse pattern in DNA methylation, i.e., low DNA methylation in ER-negative compared to ER positive patients. Therefore, the regions associated with CpG-Cluster 2B may be bound by other TFs important for ER-negative breast cancer. The lack of publically available ChIP-seq data mapping genome-wide the regions bound by ER-negative TFs hindered our eventual discovery of ER-negative TF regions affected by DNA methylation. Our preliminary search for TF motifs in the regions surrounding CpGs in Cluster 2B (±200 bp) indicated that TFs part of the Ets family may bind in these genomic regions and may be drivers of this breast cancer lineage.

Our approach of integrating genome-wide DNA methylation and expression data pinpoints important regulatory regions in breast cancer. The same approach may be applied to other physiological or pathological processes to identify epigenetically regulated enhancers, TF binding regions and can promote the discovery of new TFs associated with the process of interest.

A link between inflammation and epigenetic changes in tumor cells has recently been suggested; aberrant DNA methylation can occur in cells exposed to inflammation, and the genes epigenetically deregulated correlate with risk of disease development[42–44]. Therefore, different types of activated immune cells infiltrating breast tissues may drive specific epigenetic modifications, which in turn will play a role in cancer cell homeostasis. We found that the expression of the genes and the methylation of CpGs of Cluster 1 correlated with the level of lymphocyte infiltration in the tumor. Infiltration of immune cells has been associated with different prognosis and response to treatment independently of breast cancer subtypes[45, 46]. Our emQTL analysis represents an interesting approach to identify how the inflammatory tumor environment can affect the epigenome of the breast tumor cells. A more detailed analysis of the

DNA methylation affected by lymphocyte infiltration in Cluster 1 is an interesting avenue to explore the crosstalk between immune and tumor cells *via* the epigenome.

Through genome-wide integration of DNA methylation and gene expression data, our emQTL analysis successfully identified two gene regulatory networks altered by DNA methylation. The level of methylation at enhancers with transcription factor binding regions for ERα, FOXA1 and GATA3 appeared central in the regulation of TF target genes involved in tumor response to estrogen. Our study highlights the role of DNA methylation at these regulatory regions in giving rise to different breast cancer phenotypes.

## Methods

**Patient material.** Molecular data from two patient cohorts were publicly available and used in this study. The MicMa breast cancer cohort was collected in Oslo, Norway and has been previously described[47]. Informed consent has been obtained from all participants and the study was approved by the local ethical committee (S-97103). The DNA methylation data from this cohort was generated using the Illumina HumanMethylation450 as previously described[5], and is available in GEO with accession number GSE60185. The gene expression data was generated using Agilent whole genome 4 × 44 K oligo array as previously described[48], and is available in GEO with accession number GSE19783.

The breast cancer cohort of TCGA has been described previously[49], and the DNA methylation (level 3) and gene expression data (level 3) were downloaded from the TCGA Data Portal (https://tcga-data.nci.nih.gov). For both data types, probes with more than 50% missing values were removed, and further missing values were imputed using the function pamr.knnimpute (R package pamr) with $k = 10$.

The OSL2 breast cancer cohort[19] is a consecutive study collecting material from breast cancer patients with primary operable disease in several hospitals in south eastern Norway. Inclusion of patients started in 2006 and is still on-going. The study was approved by the Norwegian Regional Committee for Medical Research Ethics (approval number 1.2006.1607, amendment 1.2007.1125), and patients have given written consent for the use of material for research purposes. All experimental methods performed are in compliance with the Helsinki Declaration. Tumor tissue was cut into pieces and mixed before distribution to RNA and DNA extraction. Following this procedure, RNA and DNA originate from the same tissue composition. DNA from tumor tissue was isolated using the Maxwell ® 16 (Promega) instrument and the Maxwell® 16 tissue DNA Purification Kit (Promega). The DNA was eluted in 200–600 μl TE buffer (pH 8.5) and stored at −20 C. The mRNA expression data and PAM50 classification from the OSL2 cohort are available in GEO with accession number GSE58215[19].

**DNA methylation analysis.** The DNA methylation levels of more than 450,000 CpG sites were interrogated for 330 patient tumors from the OSL2 cohort using Illumina Infinium HumanMethylation450 microarray as previously described[5, 50]. Preprocessing and normalization involved steps of probe filtering, color bias correction, background subtraction and subset quantile normalization as previously described. The DNA methylation data (both raw data and preprocessed data) is available in the GEO database as a Series record with accession number GSE84207.

**Statistical and bioinformatic analyses.** All analyses were performed in the R software[51] unless otherwise specified.

**Genome-wide correlation analysis.** Correlation between the level of DNA methylation of single CpGs and gene expressions was tested assuming as null hypothesis zero correlation against non-zero correlation, using the Pearson correlation statistics (function eMap1, R package eMap)[52]. 189,026 CpGs with interquartile range of methylation values above 0.1, were included in the analysis. In the discovery cohort (MicMa) the correlation of the DNA methylation of these CpGs was tested for all gene expressions (17,558), resulting in more than three billion tests. An association was considered significant if a Bonferroni corrected *p*-value was below 0.05 (nominal *p*-value < 1.51e−11). Validation was performed by reanalyzing the significant associations from the discovery cohort in the validation cohort (TCGA); associations were confirmed when the Bonferroni corrected *p*-value was below 0.05 (nominal *p*-value < 6.12e−8). Only confirmed associations were included in further analyses.

**Hierarchical clustering of emQTL.** Hierarchical clustering was performed on the significant *p*-values from the genome-wide correlation analysis. Genes and CpGs with at least one significant association were included in the analysis. The CpG-CpG distance matrix and gene-gene distance matrix were computed using Pearson correlation (function cor) and the clustering was performed with average linkage (function hclust).

**Hierarchical clustering of methylation and gene expression.** Hierarchical clustering of CpG DNA methylation or gene expression was performed using the R package pheatmap[53] with Euclidean distance and average linkage.

**Circos plots.** Plots to visualize epigenomic connections were generated using Circos version 0.67-7[54].

**Motif enrichment analysis.** TF motif enrichment analysis was performed using the findMotifsGenome.pl script from HOMER V4.7.2 (http://homer.salk.edu/homer/chipseq/) with default parameters[55]. Motif enrichment analysis was performed ± 200 bp around CpGs.

**ChIP-seq peaks and ChIA-PET loops enrichment analysis.** We obtained the ChIP-seq peak regions in the narrowPeak format for the 689 uniformly processed ENCODE ChIP-seq experiments from the UCSC genome browser at http://hgdownload.cse.ucsc.edu/goldenPath/hg19/encodeDCC/wgEncodeAwgTfbsUniform/ (as of April 12th 2013). The enrichment between regions surrounding (± 200 bp) CpGs in Cluster 1 or Cluster 2 and ChIP-seq peak regions for each experiment was computed using the mergePeaks tool from HOMER. For the specific analyses of MCF7 TF ChIP-seq data sets, we retrieved hg19 ENCODE ChIP-seq peak regions in the bed narrowPeak format from the ENCODE data portal (https://www.encodeproject.org) and ChIP-seq peak regions from GEO from the ReMap catalog (http://tagc.univ-mrs.fr/remap/)[30]. Overlaps between CpGs in emQTL, Cluster 1, and Cluster 2 and ChIP-seq peak data sets were obtained using the bedr R package v.1.0.3 (https://CRAN.R-project.org/package=bedr). We assessed the enrichment using the hypergeometric tests (phyper R function) with all Illumina Infinium HumanMethylation450 BeadChip CpGs as background (Supplementary Data 1). Supplementary Fig. 2B was constructed using data sets showing a hypergeometric $\log_{10}$ *p*-value < −100 with the Intervene tool[56] to obtain a SVG figure further edited for colors using inkscape (https://inkscape.org/en/). We retrieved ChIA-PET pol2 loops in the MCF7 cell lines from ENCODE[57], an emQTL (CpG-Gene pair) was considered to be in a ChIA-PET loop if the CpG and the TSS of the gene were in corresponding genomic interval defining Pol2 loops. Enrichment was calculated using the hypergeometric tests (phyper R function) with all Illumina Infinium HumanMethylation450 BeadChip CpGs as background.

**ChIP-seq data reprocessing and analysis.** ChIP-seq data were acquired from publicly available sources. The accession numbers are as follow: ChIP-seq of ESR1 (GSM798423, GSM798424, GSM798425), FOXA1 (GSM986065 and carroll-lab.org.uk/FreshFiles/Data/Data_Sheet_3/MCF7_FOXA1%20binding.bed), GATA3 (GSM986068, GSM986070, GSM986072). If already aligned bed files and/or called peaks using MACS were not available: data were reprocessed. Raw reads were aligned to human reference genome (hg19) with Novoalign v2.08.02 default parameters (http://www.novocraft.com/products/novoalign/). Reads with quality ≤ 20 were considered as low quality and excluded from further analyses. Peaks associated with a specific TF were identified versus respective input background data set using MACS 1.4.1 20110622 and default parameters[58]. In further analysis a CpG was considered to be in a TF binding regions if it was found in any of the peaks associated with the ChIP-seq of this TF.

**Genomic segmentation and annotation.** The ChromHMM segmentation of the MCF7 genome was obtained from Taberlay et al.[21]. Based on ChIP-seq data of key histone modifications (H3K4me1, H3K27ac, H3K4me3, H3K27me3) and regulatory factors (CTCF, RNAPol II), a multivariate hidden Markov model annotate the MCF7 genome into nine distinct chromatin states: heterochromatin, repressed, transcribed, enhancers, enhancers + CTCF, CTCF, promoters + CTCF, promoters and promoter_poised. However, in downstream analysis some annotations were collapsed into one as follow: Enhancer = 'Enhancer' and 'Enhancer + CTCF' and Promoter = 'Promoter', 'Promoter + CTCF' and 'Poised Promoter'.

**Gene set enrichment analysis.** Gene set enrichment analysis was performed using the Molecular Signatures Database v4.0 (MSigDB;[20]) H and C5 collections. Enrichment was assessed by hypergeometric testing as implemented in the R stats package.

**MCF7 culture.** The breast cancer cell line MCF7 was obtained from American Type Culture Collection (ATCC, Manassas, VA). The cells were plated and grown for 24 h in DMEM containing phenol red and supplemented with 10% serum, 2 mM L-glutamine, 50 U/ml penicillin and 50 μg/ml streptomycin (all from Life Technologies GmbH). For hormone deprivation experiments, cells were grown for three days in DMEM without phenol red (Life Technologies GmbH) and supplemented with 5% charcoal stripped heat-inactivated FBS (HyClone), 2 mM l-glutamine, 50 U/ml penicillin and 50 μg/ml streptomycin. At day three, cells were stimulated with vehicle (ethanol) or 100 nM estradiol (Sigma–Aldrich).

**Gro-seq analysis**. Global run-on and library preparation for sequencing was performed as previously described in[59] with minor modifications. Nuclei isolation was performed 40 min after stimulation and $5 \times 10^6$ nuclei were used for each run-on reaction, 2 biological replicates were produced for both vehicle and estrogen treatments. Br-UTP was incorporated into on-going transcription by run-on reaction which was performed at 30 degree for 5 min. Total RNA was extracted with TRIzol Reagent (Life Technologies) and fragmented with RNA Fragmentation Reagent (Life Technologies). Fragmented RNA was purified with P-30 column (Bio-Rad, Hercules, CA, USA), which was followed by T4 polynucleotide kinase (PNK; New England Biolabs) treatment to dephosphorylate the 3′ end of RNA fragments. Br-UTP labeled RNA was enriched twice with anti-BrdU beads (Santa Cruz Biotech) and precipitated overnight. PolyA tailing was done using *E.coli* Poly (A) Polymerase (New England Biolabs), followed by reverse transcription with oNTI-223-index: /5Phos/5′-GATCGTCGGACTGTAGAACTCTGAAC/iSp18/ TCAGACGTGTGCTCTTCCGATCTTTTTTTTTTTTTTTTTTTTVN-3′, which allows custom barcoding. Exonuclease I (New England Biolabs) was used to remove excess oligo after reverse transcription. DNA-RNA duplex was purified with ChIP DNA Clean & Concentrator Kit (Zymo Research Corporation) followed by RNAse H treatment. cDNA was circularized with Circligase II (Epicenter) and amplified with oNTI-201: 5′-AATGATACGGCGACCACCGACAGGTTCA-GAGTTCTACAGTCCGACG-3′ and oNTI-200: 5′-CAAGCAGAAGACGGCA-TACGAGATXXXXXXGTGACTGGAGTTCAGACGTGTGCTCTTCCGATCT-3′ (XXXXXX is barcode used for specific sample) for 12–14 cycles. Final PCR product was purified by running 10% TBE gel and cleaned up. Libraries were sequenced on Illumina Genome Analyzer HiSeq 2000 according to the manufacturer's instructions.

**GRO-seq data processing and data analysis**. GRO-seq data was first trimmed from the 3′-end to remove PolyA tail with homerTools (Homer 4.8) followed by quality filtering with FASTX Toolkit (minimum 97% of base pairs should have quality scores higher than 10). Trimmed and quality-filtered data was aligned to human genome hg19 with Bowtie. Differential expression analysis was performed with edgeR, thresholds for analysis were $p$-value $< 0.05$, RPKM $> 0.5$ and log fold change $>1$ or $<-1$.

**siRNA analysis**. All siRNA experiments were at a final concentration of 50 nM. Transfections were conducted using Lipofectamine 2000 (Invitrogen). For the microarray expression experiment we used the siRNAs directed against *FOXA1* against non-targeting siRNA. The microarray data from *GATA3* siRNA experiments were available at GSE39623[16].

**Inhibition of ERα binding at enhancer**. pdCas9-PuroR plasmid expressing both dCAS9 and sgRNA was purchased from Addgene (Addgene number #71667). sgRNAs targeting the selected enhancer identified through emQTL analysis were designed by using the CRISPR Design webtool: http://crispr.mit.edu/ (sequence: TTGGAGTTGACCTCATTCCAAGG). sgRNA was synthesized by oligo annealing and cloned into expressing vectors via BbsI sites using BpiI/BbsI (Thermo Scientific, Waltham, Massachusetts, USA) and T4 DNA ligase (NEB, Ipswich, Massachusetts, USA). Plasmids expressing both dCAS9 and sgRNAs were transfected into MCF7 cells with Lipofectamine 3000 (Thermo Scientific, Waltham, Massachusetts, USA) following the manufacturer´s protocol. For gene expression analysis, cells were selected with puromycin (1 µg/ml) for 48 h after 48 h of transfection. Analysis of dCAS9 binding was performed using ChIP protocol as described previously[60]. Cells were fixed with 1% formaldehyde for 10 min and then quenched with 125 mM of glycine. Cells pellets were washed with PBS and lysed with lysis buffer. DNA fragmentation was done by using Bioruptor sonicator (Diagenode) (15–20 cycles on and off 30 s). Chromatin was incubated overnight at 4 °C with ChIP-grade CAS9 or ERα antibodies (Active Motif, 61757 for CAS9; Santa Cruz Biotechnology, Inc, sc-543x for ERα). 5 µg of each antibody per ChIP assay (1:300 dilution in volume) and equal amounts of ChIP-Grade Protein A&G Agarose Beads were added (Life technologies). The beads were washed and DNA was eluted by reverse crosslinks overnight at 65 °C. Enriched target sequences were analyzed with qPCR with the primer pair: PGR_enh_1_F: 5′-CATTCTGGGAC-TAGGCCAGC-3′ and PGR_enh_1_R: 5′-ATTCCAAGGCAGAGCTCAGG-3′

**In silico nanodissection**. The algorithm Nanodissect (http://nano.princeton.edu[34]) was used for prediction of lymphocyte infiltration. The breast collection data (May 2013), which contains 17940 genes measured on 622 arrays, was inspected for genes specifically expressed in lymphocytes (standard genes; $n = 476$; available online and defined from expert literature review) and not expressed in mammary gland ($n = 777$) or mammary epithelium ($n = 79$). The genes with more than 65% probability to be positive lymphocyte-specific standard genes as opposed to mammary gland or epithelium were further used in downstream analysis to score each TCGA and OSL2 samples for the level of lymphocyte infiltration. The average expression of the set of standard genes in a sample reflected lymphocyte infiltration and was used to divide samples of the cohorts in four groups according to quartiles.

**Pathological assessment of lymphocyte infiltration**. Vascular invasion, inflammatory cell infiltrate and necrosis, including relation of tumor cells/tumor stroma, were evaluated on slides stained with hematoxylin and eosin as previously described[61]. Using a simple microscope, subjective categorization of inflammatory cell infiltrate into the categories of "low", "intermediate-low", "intermediate-high" and "high" was performed based on the frequency of mononuclear inflammatory cell infiltration observed in invasive the tumor.

**Data availability**. The DNA methylation data from the MicMa breast tumor cohort and normal breast tissue from reduction mammoplasty is available in GEO with accession number GSE60185. The gene expression data from the MicMa cohort is available with accession number GSE19783. The newly generated DNA methylation data for the breast cancer cohort OSL2 ($n = 330$) is available in GEO with accession number GSE84207, and the mRNA expression data from the OSL2 cohort is available in GEO with accession number GSE58215. DNA methylation several cell lines were used in this study and are available in GEO: breast cancer cell lines (GSE94943), T cells (GSE79144), B cells (GSE68456), leukocytes (GSE69270) and monocytes (GSE68456). The microarray data from GATA3 siRNA experiments are available in GEO with accession number GSE39623. The ChIA-PET Pol2 data is available through the ENCODE project with accession number ENCSR000CAA. MCF7 GRO-seq data are available at GEO with the accession number GSE99508.

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

## Acknowledgements

This study was supported by funding from the KG Jebsen Centre for Breast Cancer Research (SKGJ-MED-004), and the South Eastern Norway Health Authority (grant 2011042 to VN Kristensen). Expression profiling was performed with funding from the Research Council of Norway (grant 193387/H10 to A-L Børresen-Dale and VN Kristensen), South Eastern Norway Health Authority (grant 39346 to A-L Børresen-Dale) and the Norwegian Cancer Society. T.F. and X.T. are postdoc fellows funded by the Norwegian cancer Society (grant nr 419616111190). R.E. was funded by https://www.forskningsradet.no/ (231217/F20) and https://kreftforeningen.no (3485238-2013). A.M. was supported by funding from the Norwegian Research Council, Helse Sor-Ost, and the University of Oslo through the Centre for Molecular Medicine Norway (NCMM), which is a part of the Nordic European Molecular Biology Laboratory partnership for Molecular Medicine. We would like to acknowledge Grethe I.G. Alnæs, Jovana Klajic and Eldri Undlien Due for assisting in array analyses and DNA and RNA extraction, and Georgios Magklaras for systems support.

## Author contributions

T.F.: initial discovery, designed the study, performed analysis, wrote and revised the manuscript. X.T.: designed the study, performed analysis, wrote and revised the manuscript. A.M.: bioinformatics analysis of motif and ChIP-seq enrichments, revised the manuscript. S.W.: performed functional experiments, revised the manuscript. D.N.: bioinformatics analysis, revised the manuscript. H.P.D.: scoring of FFPE for inflammation, revised the manuscript. K.K.S.: organized OSBREAC study, validation cohort, revised the manuscript. E.S.: provided tissue samples, validation cohort, revised the manuscript. A.-L. B.-D.: organized OSBREAC study, contributed funding, revised the manuscript. EB: tissue samples pathology, revised the manuscript. B.N.: provided tissue samples, discovery cohort, revised the manuscript. R.E.: provided critical points of view, revised the manuscript. A.F.: supervised statistical analysis, revised the manuscript. J.T.: normalization of DNA methylation data, revised the manuscript. A.H.: designed functional experiments and discussed the design of the study, revised the manuscript. V.N.K.: designed and funded the study, revised the manuscript, supervised all steps of the study

## Additional information

**Competing interests:** The authors declare no competing financial interests.

## Oslo Breast Cancer Research Consortium (OSBREAC)

Torill Sauer[12], Jürgen Geisler[13,14], Solveig Hofvind[15,16], Tone F Bathen[17], Olav Engebraaten[6,11,18], Øystein Garred[4], Gry Aarum Geitvik[1], Anita Langerød[1], Rolf Kåresen[6,11], Gunhild Mari Mælandsmo[18,19], Hege G Russnes[1,4], Therese Sørlie[1], Ole Christian Lingjærde[20,21], Helle Kristine Skjerven[22], Daehoon Park[23] & Britt Fritzman[24]

[12] Department of Pathology, Institute of Clinical Medicine, Akershus University Hospital, Lørenskog, Norway. [13] Department of Oncology, Institute for Clinical Medicine, Akershus University Hospital, Lørenskog, Norway. [14] Division of Medicine, Akershus University Hospital, Lørenskog, Norway. [15] Cancer Registry of Norway, Oslo, Norway. [16] Oslo and Akershus University College of Applied Sciences, Faculty of Health Science, Oslo, Norway. [17] Department of Circulation and Medical Imaging, Norwegian University of Science and Technology (NTNU), Trondheim, Norway. [18] Department of Tumor Biology, Institute for Cancer Research, Oslo University Hospital, Oslo, Norway. [19] Department of Pharmacy, Faculty of Health Sciences, University of Tromsø, Tromsø, Norway. [20] Centre for Cancer Biomedicine, University of Oslo, Oslo, Norway. [21] Department of Computer Science, University of Oslo, Oslo, Norway. [22] Department of Breast and Endocrine Surgery, Vestre Viken Hospital Trust, Drammen, Norway. [23] Department of Pathology, Vestre Viken Hospital Trust, Drammen, Norway. [24] Østfold Hospital, Østfold, Norway

