## [Peer Review File · Nature Communications]

Reviewers' comments:

Reviewer #1 (Remarks to the Author):

The authors computed pair-wise correlation between DNA methylation at CpGs and gene expression and identified two distinct bi-clusters of CpGs and genes. The authors showed that genes in cluster 1 were highly enriched with "immune" genes whereas genes in cluster2 were enriched with estrogen response genes. The authors further showed that the CpG loci in cluster 2 were significantly enriched with FOXA1, ERa, and GATA3 binding sites. Using the CpGs in cluster2, the authors found that differential methylation at those CpGs can distinguish ER positive from ER negative breast cancers. The authors largely validated their findings in a validation set. The authors also validated the target genes of ERa, FOXA1 or GATA3 by knocking down those factors in the MCF7 cell line.

I believe that much of the findings presented in the manuscript about breast cancer is not new. The regulatory network involving ERa, FOXA1 and GATA3 and DNA methylation at enhancer sites and the finding that DNA methylation at enhancers distinguishes distinct breast cancer subtypes may constitute a novel finding (but I am not certain).

Major comments

1) The authors used only 213 breast cancer samples from TCGA based on the availability of PAM50 information. I believe that there are many more samples with clinical information than the 213 used.

2) The association between DNA methylation and gene expression was computed using all samples. Could the observed correlation be due to the differences among tumor subtypes, e.g., ER positive, HER2-amplified, and basal-like? I think it would be helpful to repeat some of the analyses on matched RNA-seq and DNA methylation data from TCGA breast cancer datasets for each tumor subtype separately.

Minor comments

3) For the association analysis, the authors tested if the correlation coefficient between DNA methylation and gene expression is non-zero. This seems to be a very lenient. Similarly, the authors only eliminated those CpGs whose interquartile range was less than 0.1 which is quite small. CpGs with missing values in more than 50% of the samples were eliminated. Those missing values that remained were imputed. This choice also seems to be quite lenient. I guess the authors wanted to include as many genes/CpGs as possible during the discovery phase. The question is how sensitive are the conclusions to those choices?

4) The authors identified significantly enriched ERa, FOXA1, and GATA3 binding sites in cluster 2 CpG loci. Those would be expected. I would also expect enrichment for P300, ZNF217, and c-Myc in those CpG loci in the MCF7 cell line as well. ChIP-seq data for those factors in MCF7/T47D cell line are all available from ENCODE.

5) The immune cluster is very interesting. The authors attempted to explore its relationship with immune cell infiltrating. The challenge is that the CpG signature derived from tumor samples which usually consisted of tumor cells, stromal cells, infiltrating immune cells, extracellular matrix and blood vessels. There are computational methods that try to tease out the signatures for some of the components. It is not easy. The conclusion from the authors may be true but is hard to verify.

Reviewer #2 (Remarks to the Author):

The paper addresses the question of how genome wide alterations in DNA methylation patterns

could lead to breast cancer heterogeneity. This has been the subject of a number of recent papers in the literature. The unique aspect of this work and the contribution of the authors is that it is a comprehensive analysis of which CpG methylation lead to changes in the transcriptome and are linked to transformation by using a genome wide expression-methylation quantitative trait loci (emQTL) analysis between 5MeCpG and gene expression.

1. They use data from two public databases (n=104 and n=253) and conduct a third analysis of DNA methylation alone on 330 samples as a validation set. Unsupervised clustering of Bonferroni corrected p-values identified two strong clusters of CpGs and genes- Cluster 1 associated with immune cell homeostasis, while Cluster 2 contained motifs associated with FOXA1 and GATA3. The high degree of correlation between methylation and expression of CpGs and genes in the two clusters is impressive.

2. The authors conclude that CpGs captured by emQTL were enriched at enhancers and specific TF binding regions. How many new sites were identified for each cluster, how far from the CpG, and what is the location of these enhancers? It appears that 2/3rd of the Cluster 2 CpGs were significantly enriched at ER α , FOXA1 and GATA3 binding regions. What about the 1/3rd of Cluster 1 sites enriched at the same ER α , FOXA1 and GATA3 binding regions? What function do they perform in those tumors? It will be informative to see ESR1, GATA3, FOXA1 enrichment in cluster 1, the same way that is displayed Figure 2B. Perhaps the authors have addressed this: What is the degree of ESR1, GATA3, FOXA1 binding site enrichment for randomly sampled CpGs in the genome?

3. How does methylation at enhancers and transcription factor binding sites affect breast cancer pathogenesis? The authors ask how the level of methylation in CpGs in Cluster 2 correlate with histopathological features and molecular classification of breast cancer patients. Unsupervised clustering of DNA methylation for 213 breast cancer sample with available PAM50 information were analyzed. Were both pieces of information available for only 213 tumors? Was this analyzed for just a subset of the total tumors? Along those lines, what was their sample selection criteria? Be that as it may, the data is impressive- a clear distinction between Cluster 1 and Cluster 2 is evident, CpGs in Cluster 2 subdivide into 2A and 2B with lower and higher methylation in ER α , FOXA1 and GATA3 binding regions in ER+ tumors. In their discussion, the authors do not mention the ER+/HER2 positive group. Does this second subgroup relate to the luminal B subtype of tumors? Since ER α , FOXA1 and GATA3 are well known to be intimately associated with the pathogenesis of ER+ tumors, it would have been informative to investigate some other pathway as well associated CpG/expression correlations, so as to reveal novel mechanisms involved in ER+ breast cancer.

4. Functional validation was performed by identifying targets of FOXA1 and GATA3 following knockdown of FOXA1 or GATA 3 by siRNA in MCF7 cells. This analysis does not verify the function of the putative enhancers newly identified by the authors. A stronger direct correlation should be sought by performing ChIP analysis to determine FOXA1 or GATA 3 occupancy at the site that also leads to a change in the expression of the target gene.

Other comments:

- Supp figure 1 is helpful to the reader to understand the flow of information. Please include the 3rd cohort information in this flowchart. No data is provided except that the findings were validated. Expand on these findings.
- Figure 2 heading is incorrect. I believe this is expression data, not methylation.
- Figure 4: Differential expression of genes in Cluster 2. A and B) Unsupervised clustering of expression of the 259 genes in Cluster 2 from A) the 272 samples in the OSL2 cohort with PAM50 are mirror images of each other in their organization. Arranging the PAM50 reds vs blues in the same order would be helpful to the reader.
- Figure 5 G-I : looks only at 1 CpG site and correlation to expression. Is it possible to look at all the CpGs in promoter/first exon and their relationship to TF expression? Why was this site alone chosen?
- Change the colors for Immune Infiltration, Int_high and High - too similar so differences, especially in the strips on heatmaps are not visible.
- Describe in the figure legend that Fig 2 uses HM450k and S. Fig 2 uses hg19 annotation; it took me a while to find the differences since it is embedded in the text.

We are thankful for the many positive comments and constructive suggestions made by the reviewers. In our belief, the additional analysis required by the reviewers strengthens the conclusion, all changes in the Manuscript are highlighted in blue.

Reviewers' comments:

Reviewer #1 (Remarks to the Author):

The authors computed pair-wise correlation between DNA methylation at CpGs and gene expression and identified two distinct bi-clusters of CpGs and genes. The authors showed that genes in cluster 1 were highly enriched with “immune” genes whereas genes in cluster2 were enriched with estrogen response genes. The authors further showed that the CpG loci in cluster 2 were significantly enriched with FOXA1, ERa, and GATA3 binding sites. Using the CpGs in cluster2, the authors found that differential methylation at those CpGs can distinguish ER positive from ER negative breast cancers. The authors largely validated their findings in a validation set. The authors also validated the target genes of ERa, FOXA1 or GATA3 by knocking down those factors in the MCF7 cell line.

I believe that much of the findings presented in the manuscript about breast cancer is not new. The regulatory network involving ERa, FOXA1 and GATA3 and DNA methylation at enhancer sites and the finding that DNA methylation at enhancers distinguishes distinct breast cancer subtypes may constitute a novel finding (but I am not certain).

We agree with the reviewer that the pivotal roles of ER α , FOXA1, and GATA3 have been known and extensively studied in breast cancer. What came as a surprise to us and what we believe is an important novel finding is how heavily epigenetic regulation dominated this transcriptional network and how distinctly sufficient it was for tumor classification. These results emerged from our fully unsupervised analysis of all possible associations between DNA methylation and gene expression *in cis* and *in trans*, and as the first evidence that *in trans* associations are able to identify distinct transcriptional networks with biological significance. By rephrasing some paragraphs and adding data to the manuscript, we hope that we make this clearer and that the reviewer will agree that our data is novel and important for the breast cancer field, and also that the method we present may have far-reaching implications for identifying connections between the epigenome, transcription factor activity and physiological and pathological states in other diseases and conditions.

Major comments

1) The authors used only 213 breast cancer samples from TCGA based on the availability of

PAM50 information. I believe that there are many more samples with clinical information than the 213 used.

In the emQTL validation of the MicMa we use all TCGA samples for which there are both DNA methylation and gene expression data performed by microarray. This corresponds to 253 samples as indicated in the pipeline of analysis shown in Supplementary Figure 1.

Of these 253 samples, 213 had available PAM50 subtyping. In the first version of the manuscript, we used the methylation data from these 213 to performed unsupervised clustering. As the reviewer suggested, we redid our analysis maximizing the usage of TCGA samples; whenever we did not need matching DNA methylation and expression data, we increased the number of samples used for analysis (i.e. DNA methylation and gene expression heatmaps). The methylation heatmap of TCGA samples now include 609 samples, and the gene expression heatmap includes 528 patients. When samples had missing PAM50 information, we used the *genefu* R package for subtyping. Updated heatmaps are found in Figure 3A, 4B and 8A, corresponding sections of the materials and methods and Figure legends have been updated. The interpretations of the methylation and gene expression heatmaps did not change, and the results are stronger with more samples. We thank the reviewer for pointing this out.

2) The association between DNA methylation and gene expression was computed using all samples. Could the observed correlation be due to the differences among tumor subtypes, e.g., ER positive, HER2-amplified, and basal-like? I think it would be helpful to repeat some of the analyses on matched RNA-seq and DNA methylation data from TCGA breast cancer datasets for each tumor subtype separately.

Throughout the analyses, we used the microarray-based gene expression data to allow more comparable results between the discovery and validation cohorts.

We agree with the reviewer that the associations we observe may be related to the heterogeneity between tumor subtypes. In fact, it is evident from our data that the emQTLs in Cluster 2 (estrogen cluster) are a result of reciprocal methylation/expression pattern in ER positive (Luminal) compared to ER negative (Basal) tumors. Indeed, the identification of Cluster 2 is strongest after inclusion of both ER positive and ER negative tumors in the analysis. A paragraph has been added to the discussion (p11-12).

To further reveal the role of inter-tumoral variation and better answer the reviewer's comment, we performed subtype specific analysis of DNA methylation and gene expression correlations within the PAM50 subtypes. To maximize the number of samples within each subtype, and avoid extensive loss of statistical power, we used the samples from TCGA with overlapping DNA methylation and RNA-seq expression data. With this approach, we obtained 236 Luminal A, 137 Luminal B, 49 Her2 enriched, 92 Basal-like, and 44 Normal-like samples. The analysis with Luminal A, Luminal B and Basal-like should have comparable or better statistical power compared to the original analysis (N=104), while the analysis for Her2 enriched and Normal-like may be underpowered.

Each subtype specific analysis was performed the same way as the original analysis of the MicMa cohort, i.e. Pearson correlation was computed between CpG methylation (IQR>0.1) and gene expression *in cis* and *in trans*, and associations were considered significant if the Bonferroni corrected p-values were smaller than 0.05. The results are shown in the table below and added as Supplementary Table 5.

Table: Subtype specific emQTL analysis. Number of significant associations, CpGs or Genes is shown, and the percentage of rediscovered associations, CpGs or genes is shown in parenthesis.

	Luminal A N=236	LuminalB N=137	Her2 enriched N=49	Basal-like N=92	Normal-like N=44
Total number of associations	4,689,557	1,647,259	21,161	2,036,164	178
emQTL rediscovered	244,388 (33.0%)	149,841 (20.3%)	4,212 (0.57%)	116,206 (15.7%)	17 (0.00023%)
Cluster 1 genes rediscovered	159 (98.8%)	159 (98.8%)	96 (59.6%)	153 (95,0%)	16 (9.9%)
Cluster 1 CpGs rediscovered	2,471 (72.7%)	2,070 (60.9%)	1,082 (31.8%)	2,389 (70.2%)	4 (0.12%)
Cluster 2 genes rediscovered	137 (50.6%)	115 (42.4%)	6 (2.2%)	14 (5.2%)	6 (2.2%)
Cluster 2 CpGs rediscovered	2,514 (69.8%)	1,624 (45.1%)	74 (2.1%)	107 (3%)	13 (0.36%)

The number of identified associations varies greatly and depends on the number of samples included in the analysis. For Luminal A, Luminal B and Basal-like we identified more than one million significant associations, while the analyses of Her2 enriched and Normal-like identified relatively few associations likely due to low statistical power. Strikingly, we rediscover Cluster 1 in all subtypes (except Normal-like), identifying more than 95% of the Cluster1-genes for Luminal A, Luminal B and Basal-like. Cluster 2 was to a lesser degree rediscovered, identifying approximately 50% of the Cluster2-genes in Luminal A and Luminal B and very few in the other subtypes. These data are described in the result section (p11-12)

As we also argue in other parts of the article, Cluster1 is related to immune infiltration and intratumor heterogeneity, is therefore independent of subtypes and rediscovered in subtype specific analyses. Conversely, Cluster 2 is more related to inter-tumor heterogeneity (ER positive versus ER negative tumors), and this cluster can be fully identified when all subtypes are included. These observations are described in the discussion section (Discussion p 12).

Minor comments

3) For the association analysis, the authors tested if the correlation coefficient between DNA methylation and gene expression is non-zero. This seems to be a very lenient. Similarly, the authors only eliminated those CpGs whose interquartile range was less than 0.1 which is quite small. CpGs with missing values in more than 50% of the samples were eliminated. Those missing values that remained were imputed. This choice also seems to be quite lenient. I guess the authors wanted to include as many genes/CpGs as possible during the discovery phase. The question is how sensitive are the conclusions to those choices?

All Pearson correlation tests were performed with the null hypothesis that there was no correlation between methylation and expression and the alternative hypothesis that there was a correlation between methylation and expression (non-zero), with a direct measure of the correlation coefficient and significance, which was Bonferroni corrected. In order to

encompass all possible correlations, we performed the analysis trying to be most including in the input and most conservative in the output, relying on Bonferroni correction and repeated validation for the final estimate of significance.

In the output, we do not have any restrictions on correlation coefficient; but the p-value cutoffs caused by Bonferroni correction are extremely strict ($1.5e-11$ in discovery and $6.1e-8$ in TCGA validation). The p-value cutoffs have now been specifically described in the Materials and Methods (p 16). Despite these strict p-values cutoff, our results were consistent across three independent patient cohorts: 1) the original MicMa ($n=104$); 2) TCGA ($n=253$); and 3) OSL2 ($n=277$), this has been described in the Result section (p 4).

Concerning the input of CpGs, the cutoff of $IQR > 0.1$ was chosen to balance false negatives (caused by overly strict multiple testing correction) and loss of CpGs from the analysis. There are many CpGs whose DNA methylation do not change in breast cancer and these were removed by the $IQR > 0.1$ to avoid unnecessary tests. On the other hand, there are CpGs with IQRs between 0.1 and 0.2 that show consistently differential methylation between certain groups and these we wanted to keep in the analysis.

To formally answer the reviewer's comment on how the emQTL may be influenced by cutoff parameters, we tested the influence of cutoffs and performed an analysis on the OSL2 choosing CpGs with $IQR > 0.2$ and CpG-gene associations with absolute correlation > 0.4 . When increasing the IQR cutoff from 0.1 to 0.2 fewer CpGs were included in the analysis (64,059 compared to 182,620). Further, the CpG-gene associations were restricted to those with absolute correlation > 0.4 . Overall, 189,816 of the original 739,608 emQTL were rediscovered. Importantly, 97% of the genes in Cluster 1 and 95% of the genes in Cluster 2 were rediscovered using these new parameters. Also, when performing p-value clustering on these results, Cluster 1 and Cluster 2 were also identified (shown in the figure below.)

In summary, both Cluster 1 and Cluster 2 were retained using different cutoffs. This further supports that our conclusions are not the result of choices of CpGs and statistical cutoff.

4) The authors identified significantly enriched ERα, FOXA1, and GATA3 binding sites in cluster 2 CpG loci. Those would be expected. I would also expect enrichment for P300, ZNF217, and in those CpG loci in the MCF7 cell line as well. ChIP-seq data for those factors in MCF7/T47D cell line are all available from ENCODE.

Our choice of ChIP-seq data and transcription factors to analyze through our manuscript (ESR1, FOXA1 and GATA3) and further functional studies was based on motif analysis (Supplementary Table 1B) and screening of experimentally defined TF binding regions using the 689 uniformly processed human ChIP-seq data sets from ENCODE (Supplementary Table 2B). Both analyses showed that binding regions for ESR1, FOXA1 and GATA3 were enriched in the vicinity of CpGs in Cluster 2. Actually, we were expecting to see many TFs enriched in Cluster 2 and were surprised by the strong dominance of enrichment of ESR1, GATA3 and FOXA1 which dominance we consider to be one of the main messages of the paper.

To double check that we were not missing other TFs and to answer the reviewer's comment we further investigated the transcriptional network involved in Cluster 2. We analyzed in greater detail 71 available ChIP-seq datasets from the MCF7 cell line available at ENCODE and 40 available at GEO. We performed enrichment analysis as suggested by the reviewer. ESR1, FOXA1 and GATA3 were the most enriched TFs identified in Cluster 2-CpGs. As the reviewer envisioned, we also found GREB1, FOS, DPFB, AHR and ZNF217 enriched to a lesser extent. We have added these results to Supplementary Table 3 and Supplementary Figure 2B and presented them in the manuscript (p 6)

5) The immune cluster is very interesting. The authors attempted to explore its relationship with immune cell infiltrating. The challenge is that the CpG signature derived from tumor samples which usually consisted of tumor cells, stromal cells, infiltrating immune cells, extracellular matrix and blood vessels. There are computational methods that try to tease out the signatures for some of the components. It is not easy. The conclusion from the authors may be true but is hard to verify.

Our conclusion that Cluster 1 is associated with immune infiltration stems from two agreeing pieces of evidence: 1) the high correlation of the expression of Cluster 2 genes with the immune infiltration assessed by pathology and 2) the *in silico* estimated lymphocyte infiltration performed by Nanodissect (Ju W, et al. Genome research 23, 1862-1873 (2013)). Nanodissect enabled us to identify genes whose expression reflects lymphocyte infiltration in breast cancers. The mean expression of lymphocytes related genes was used to score each tumor for lymphocyte infiltration and in our manuscript we show that it strongly correlates with the pathologically assessed infiltration. We concluded that the level of expression and methylation of genes and CpGs in Cluster 1 is affected by the level of lymphocyte infiltration in the tumor.

However, we agree with the reviewer that most of our observations showing that Cluster 1 was associated with immune infiltration were based only on gene expression. Therefore, upon the reviewer's request we further characterized the methylation of the Cluster 1-CpGs.

Cluster 1-CpGs showed either positive (Cluster 1B) or negative (Cluster 1A) correlations with Cluster 1 genes. These two sub-clusters of CpGs have now been clearly identified in the heatmap (Supplementary Figure 7). We found that average DNA methylation in Cluster 1A and Cluster 1B was highly dependent of lymphocyte infiltration. In Cluster 1A, DNA methylation decreased with increasing lymphocyte infiltration (Supplementary Figure 9A), while in Cluster 1B DNA methylation increased with the level of lymphocyte infiltration (Supplementary Figure 9B).

To investigate whether the changes in DNA methylation associated with lymphocyte infiltration may be a result of a gradual mix of different DNA methylation signals arising from different cell types (breast cancer cells and lymphocytes), we (i) investigated DNA methylation of Cluster 1-CpGs in "pure" cell types (Breast cancer cell lines and immune cells) and we (ii) assessed for intratumor heterogeneity using DNA methylation data.

- i. We observed that DNA methylation of breast cancer cell lines resembled the methylation of tumors with lower infiltration. Tumors with higher lymphocyte infiltration showed intermediate methylation levels between breast cancer cell lines and immune cells. This holds true for both Cluster 1A and Cluster 1B (Supplementary Figure 9A and 9B). This supports that the observed methylation level in tumors with higher infiltration may be attributed to higher numbers of immune cells.
- ii. When using DNA methylation to assess tumor purity (Zheng et al. Genome Biology 18:17 (2017)), we found that tumors with higher lymphocyte infiltration had significantly lower tumor purity (Supplementary Figure 9C). This further supports that the DNA methylation measurements are affected by immune infiltration and intratumor heterogeneity.

As the CpGs in Cluster 1 are associated with immune cell infiltration, they may represent some interesting targets to complement or implement immune cell infiltration deconvolution using DNA methylation, which we may explore in the future.

These new results produced upon the reviewer's suggestion are now presented in the Manuscript (p9-10, of the result section) and presented in Supplementary Figure 9.

Reviewer #2 (Remarks to the Author):

The paper addresses the question of how genome wide alterations in DNA methylation patterns could lead to breast cancer heterogeneity. This has been the subject of a number of recent papers in the literature. The unique aspect of this work and the contribution of the authors is that it is a comprehensive analysis of which CpG methylation lead to changes in the transcriptome and are linked to transformation by using a genome wide expression-methylation quantitative trait loci (emQTL) analysis between 5MeCpG and gene expression.

1. They use data from two public databases (n=104 and n=253) and conduct a third analysis of DNA methylation alone on 330 samples as a validation set. Unsupervised clustering of Bonferroni corrected p-values identified two strong clusters of CpGs and genes- Cluster 1 associated with immune cell homeostasis, while Cluster 2 contained motifs associated with FOXA1 and GATA3. The high degree of correlation between methylation and expression of CpGs and genes in the two clusters is impressive.

We thank the reviewer for the positive comments.

2. The authors conclude that CpGs captured by emQTL were enriched at enhancers and specific TF binding regions. How many new sites were identified for each cluster, how far from the CpG, and what is the location of these enhancers?

We now provide the hg19 location of the ChromHMM enhancers identified in Cluster 2 and their annotation in regard to FOXA1, ESR1 or GATA3 binding regions (ChIP-seq peak), CpG-Cluster 2A or 2B and MCF7-ChromHMM segmentation. This information has been added to Supplementary Table 4 and described in the result section (p 6).

In the analysis in which we seek enrichment of emQTL-CpGs over the 689 human uniformly processed ChIP-seq data sets, we considered region ± 200 bp from CpG (Supplementary Tables 2A-B). Otherwise all over the manuscript, we considered a CpG to be at an enhancer or a TF binding region only and strictly if this CpG was in the genomic interval defined as enhancer by ChromHMM or TF binding region (ChIP-seq peak).

We use the ChromHMM MCF7 enhancers and MCF7 super-enhancers and are not directly identifying new enhancers, rather elaborating on their function and link to epigenetic regulation in breast cancer development. We believe that our analysis is one of the first to comprehensively identify enhancers in ER+ breast cancer affected by DNA methylation.

As Cluster 2A is enriched in MCF7 enhancers and FOXA1, GATA3 and ESR1 binding sites, all genomic regions identified in Cluster2A may represent conserved enhancers across ER positive disease unknown by now. This has been added to the discussion (p 13).

It appears that 2/3rd of the Cluster 2 CpGs were significantly enriched at ER α , FOXA1 and GATA3 binding regions. What about the 1/3rd of Cluster 1 sites enriched at the same ER α ,

FOXA1 and GATA3 binding regions? What function do they perform in those tumors? It will be informative to see ESR1, GATA3, FOXA1 enrichment in cluster 1, the same way that is displayed Figure 2B. Perhaps the authors have addressed this: What is the degree of ESR1, GATA3, FOXA1 binding site enrichment for randomly sampled CpGs in the genome?

We analyzed to which degree the CpGs in Cluster 1 were enriched at FOXA1, ESR1 and GATA3 binding regions and found that Cluster1-CpGs were enriched at these three TF binding regions using hypergeometric testing, however to a lower degree than Cluster2-CpGs. This data are shown in Supplementary Figure 8 and Supplementary Figure 2B.

To more specifically answer the reviewer, we performed a permutation test (1000 permutations) in which we randomly selected 3601 CpG (number of CpGs in Cluster 2) from the 450K array and calculated how many were in either ESR1 GATA3 or FOXA1 binding regions. None of the randomly selected combination of CpGs showed the same enrichment as the CpGs in Cluster2 (p-value < 0.0001). This has been added in the paper (p 6).

3. How does methylation at enhancers and transcription factor binding sites affect breast cancer pathogenesis? The authors ask how the level of methylation in CpGs in Cluster 2 correlate with histopathological features and molecular classification of breast cancer patients. Unsupervised clustering of DNA methylation for 213 breast cancer sample with available PAM50 information were analyzed. Were both pieces of information available for only 213 tumors? Was this analyzed for just a subset of the total tumors? Along those lines, what was their sample selection criteria? Be that as it may, the data is impressive- a clear distinction between Cluster 1 and Cluster 2 is evident, CpGs in Cluster 2 subdivide into 2A and 2B with lower and higher methylation in ER α , FOXA1 and GATA3 binding regions in ER+ tumors.

This question is very similar to question 1 by reviewer 1. For more details refer to our above answer. In short: in the emQTL, we had to use TCGA samples with overlapping DNA methylation and gene expression data by microarray (n=253). Whenever DNA methylation data and gene expression data are not used together, we now use as many TCGA samples as possible (DNA methylation, n=609; gene expression, n=528). See updated Figure 3A, 4B and 7A. The biological interpretation of the methylation and gene expression heatmaps did not change, and the results are stronger with more samples. We thank the reviewer for pointing this out.

In their discussion, the authors do not mention the ER+/HER2 positive group. Does this second subgroup relate to the luminal B subtype of tumors?

No specific emQTL patterns related to Her2 enriched tumor samples emerged from our initial analysis. We performed subtype specific emQTL (also as an answer to reviewer 1, question 2), and in this analysis the Her2 enriched subtype showed relatively few associations between DNA methylation and gene expression, possibly due to lack of statistical power. However, part of the immune cluster was rediscovered suggesting variable immune infiltration also within this subtype.

For the Luminal B subtype 20% of the emQTL and 99% of the emQTL genes in Cluster 1 could be rediscovered. These results are shown in Supplementary Table 5 and described in the results (p 11-12).

Since ER α , FOXA1 and GATA3 are well known to be intimately associated with the pathogenesis of ER+ tumors, it would have been informative to investigate some other pathway as well associated CpG/expression correlations, so as to reveal novel mechanisms involved in ER+ breast cancer.

Our initial choice of studying ER α , FOXA1 and GATA3 signaling pathway was based on motif analysis around the CpGs of Cluster 2 (+/- 200bp; Supplementary Table 1B) and screening of experimentally defined TF binding regions using the 689 uniformly processed human ChIP-seq data sets from ENCODE (Supplementary Tables 2B). Both analyses showed that binding regions for FOXA1 and GATA3 were enriched in the vicinity of the CpGs in Cluster2. To further answer the reviewer's comment, we retrieved all ChIP-seq data from the MCF7 cell line available at ENCODE and GEO (71 and 40 data sets respectively) and performed enrichment analysis as suggested by the reviewer. As expected, we found that ESR1, FOXA1 and GATA3 were the most enriched TFs identified in the Cluster2, and we also found ZNF217 and FOS enriched. This data are described in the Results and shown in Supplementary Table 3 and Supplementary Figure 2B. It is also presented in the manuscript (p 6).

4. Functional validation was performed by identifying targets of FOXA1 and GATA3 following knockdown of FOXA1 or GATA 3 by siRNA in MCF7 cells. This analysis does not verify the function of the putative enhancers newly identified by the authors. A stronger direct correlation should be sought by performing ChIP analysis to determine FOXA1 or GATA 3 occupancy at the site that also leads to a change in the expression of the target gene.

To assess how the enhancers identified in Cluster 2 may regulate the expression of genes involved in estrogen signaling, we mapped long range chromatin interactions using ChiA-Pet Pol2 genome wide interactions in the MCF7 cell lines. These data allow mapping any contact point in a chromosome through pol2 and identifies potential functional loops between distant enhancers and promoters. We assessed if our observed CpG genes (emQTL) association were found in Pol2 loops. In Cluster 2A (in which CpGs are found in TF binding sites), CpG-gene associations were significantly enriched at pol2 loops (Figure 5G). These data support that the enhancers identified in Cluster2 and more specifically in Cluster2A may functionally regulate the expression of the target genes. These data are now described in the result section (p 9) and, presented in Figure 6 A and B.

To further assess the impact of TF binding at enhancers in the regulation of target gene, we experimentally decreased the binding of ER α using dCas9 and a specific gRNA directing the binding of dCas9 in an enhancer containing ER α binding sites, which was located in the 5'UTR region of the *PGR* gene. This enhancer was predicted by emQTL to affect *PGR* expression. When experimentally decreasing ER α binding at this enhancer in the MCF7 cell line, we observed a decrease in PGR expression, which demonstrated the functional

importance of the enhancer identified by emQTL analysis. These data are presented in Figure 6C-E and in the result section (p 9).

Other comments:

- Supp figure 1 is helpful to the reader to understand the flow of information. Please include the 3rd cohort information in this flowchart. No data is provided except that the findings were validated. Expand on these findings.

The flow chart Supplementary Figure 1 has been updated with the information of the newly generated validation cohort OSL2 (the 3rd cohort), some additional information have also been added in the manuscript (p 4) (“Using OSL2, we validated 95.5% of the total 98% of Cluster1 and 96.3% of Cluster2 emQTL identified from the MicMa and TCGA cohorts, which demonstrated that the 5meCpG-gene pairs are conserved across breast cancer cohorts2”)

- Figure 2 heading is incorrect. I believe this is expression data, not methylation.

We apologize, but the heading we see in our Figure 2 reflects correctly the content of the figure. Hopefully, this is a mistake somewhere in the compilation of the reviewer’s document, and it will be fixed now.

- Figure 4: Differential expression of genes in Cluster 2. A and B) Unsupervised clustering of expression of the 259 genes in Cluster 2 from A) the 272 samples in the OSL2 cohort with PAM50 are mirror images of each other in their organization. Arranging the PAM50 reds vs blues in the same order would be helpful to the reader.

The heatmaps have been organized as suggested by the reviewer to improve interpretation.

- Figure 5 G-I : looks only at 1 CpG site and correlation to expression. Is it possible to look at all the CpGs in promoter/first exon and their relationship to TF expression? Why was this site alone chosen?

Please note that due to addition of new main Figures, the previously annotated Figure 5G-I have been moved to Supplementary Figure 6A-C.

It is so that in these previous Figures 5 G-I of the submitted first manuscript, only the methylation of one CpG in the vicinity of *ESR1*, *FOXA1* or *GATA3* was chosen to demonstrate the inverse correlation with the respective TF expression. We chose these CpGs because of their strongest inverse correlation and because we wanted to show that the same mode of regulation that was spread over the whole network was also valid for the key regulators. Therefore our purpose was to pinpoint the most prominent and putative region

regulating the expression of ESR1, FOXA1 or GATA3. Indeed these CpGs are often outside the common promoter regions.

We believe that with this method without a priori we find the most prominent CpG associated with the TF expression and this may be more informative than looking at DNA methylation at promoters (i.e +/- 3kb from TSS). When we look at the inverse correlation between mean methylation values of the CpGs +/- 3kb from TSS, we found weaker and probably less biologically significant association between gene expression and DNA methylation. As shown below. This has been shortly discussed in the manuscript (p 10).

- Change the colors for Immune Infiltration, Int_high and High - too similar so differences, especially in the strips on heatmaps are not visible.

The color (deep pink) from the int-high of the infiltration has been changed to green. We also modified the annotation of infiltration into low, moderate, high, severe.

- Describe in the figure legend that Fig 2 uses HM450k and S. Fig 2 uses hg19 annotation; it took me a while to find the differences since it is embedded in the text.

We changed the legend as asked by the reviewer.

REVIEWERS' COMMENTS:

Reviewer #1 (Remarks to the Author):

The authors have addressed most of my comments.

Reviewer #2 (Remarks to the Author):

The authors have carefully responded to the critique, using more samples, to arrive at the same results. Again, the findings confirm and extend those that have been previously published in a much more rigorous manner, but do not offer any new breakthroughs